# RAQ-VAE: Rate-Adaptive Vector-Quantized Variational Autoencoder

## Abstract

Vector Quantized Variational AutoEncoder (VQ-VAE) is an established technique in machine learning for learning discrete representations across various modalities. However, its scalability and applicability are limited by the need to retrain the model to adjust the codebook for different rate requirements or encoding efficiency. We introduce the Rate-Adaptive VQ-VAE (**RAQ-VAE**) framework, which addresses this challenge with two novel discrete (codebook) representation methods: a model-based approach using a clustering technique for existing pre-trained VQ-VAE models, and a data-driven approach utilizing a sequence-to-sequence (Seq2Seq) model for variable-rate codebook generation. Our experiments demonstrate that RAQ-VAE achieves effective reconstruction performance across multiple rates, often outperforming conventional fixed-rate VQ-VAE models. This work enhances the adaptability and performance of VQ-VAEs, with broad applications in data reconstruction, generation, and computer vision tasks.

## 1 Introduction

Vector quantization (VQ) (Gray, 1984) is a fundamental technique for learning discrete representations for various tasks (Krishnamurthy et al., 1990; Gong et al., 2014; Van Niekerk et al., 2020) in the field of machine learning. The Vector Quantized Variational AutoEncoder (VQ-VAE) (Van Den Oord et al., 2017; Razavi et al., 2019), which extends the encoder-decoder structure of the Variational Autoencoder (VAE) (Kingma & Welling, 2013; Rezende & Viola, 2018), introduces discrete latent representations that have proven effective across multiple domains, including computer vision (Razavi et al., 2019; Esser et al., 2021), audio (Dhariwal et al., 2020; Yang et al., 2023; Tseng et al., 2023), and speech (Kumar et al., 2019; Xing et al., 2023). These successes are attributed to the inherently discrete nature of the data in these domains, which makes VQ well suited to learning complex inference and prediction tasks.

Recent developments have further enhanced VQ-based discrete representation learning by integrating it with deep generative models, such as Generative Adversarial Networks (GANs) (Esser et al., 2021) and Denoising Diffusion Probabilistic Models (DDPMs) (Cohen et al., 2022; Gu et al., 2022; Yang et al., 2023). As VQ-VAE is integrated into these diverse generative frameworks, its utility and applicability in various tasks are becoming increasingly evident. However, even with this success, the *scalability* of the codebook-driven quantization process poses a significant challenge, further motivating our approach. With the proliferation of large datasets and the demand for real-time processing, VQ-based architectures struggle with the computational complexity associated with dynamic compression, including the need to retrain models to adjust computational loads. Consequently, addressing the scalability of the VQ process is crucial to fully realizing the potential of VQ-VAE, especially in integrating it with large-scale generative VQ models (Yu et al., 2022).

To address the issues, Li et al. (2023) proposed a method to resize the codebook without retraining the publicly available VQ models by applying hyperbolic embeddings to enhance the codebook vector with co-occurrence information and reordering the improved codebook with a Hilbert curve. Another approach to achieve more comprehensive codebook representation, the use of multi-codebook has been an ongoing challenge to achieve richer representations for different tasks (Guo et al., 2022). Malka et al. (2023) designed and learned a nested codebook based on progressive learning to support different quantization levels. Guo et al. (2023) proposed a framework for predicting codebook indexes generated from embeddings of student models using multi-codebook vector quantization

by reformulating teacher label generation as a codec problem in knowledge distillation. Recently, Huijben et al. (2024) focused on unsupervised codebook generation based on residual quantization by studying the vector quantizer itself. However, addressing these issues through multi-codebook or residual quantization generally entails substantial changes to the existing well-established structure of VQ-VAEs, or face a reduction in the resolution of the quantized feature map.

To this end, we propose a Rate-Adaptive VQ-VAE (RAQ-VAE) framework that allows discrete representation at various rates with a single VQ-VAE model. First, we propose *model-based* RAQ-VAE that can use the existing VQ-VAE model to obtain rate-adaptive VQ through a differential $k$-means clustering (DKM) (Cho et al., 2021) algorithm and its inverse functionalization without any additional parameters and retraining. Next, we present *data-driven* RAQ-VAE with Sequence-to-Sequence (Seq2Seq) (Sutskever et al., 2014) model for rate-adaptive codebook generation. The *data-driven* RAQ-VAE can achieve discrete representation at any desired rate through the Seq2Seq model and approaches or partially outperforms the separately trained conventional VQ-VAE model. Our framework addresses the challenge of needing multiple VQ-VAE models for different compression rates, especially in large-scale computer vision tasks that require high-capacity representations. Additionally, it can be seamlessly integrated into various VQ applications without requiring significant modifications to the existing VQ-VAE structure.

Our contributions are summarized as follows:

- We introduce the RAQ-VAE framework with two VQ codebook representation methods: *model-based* RAQ-VAE, utilizing an existing trained VQ-VAE model, and *data-driven* RAQ-VAE, combining Seq2Seq model with VQ-VAE architecture.

- We propose *model-based* RAQ-VAE, which adapts the codebook of a VQ-VAE model using a dynamic codebook clustering method, allowing the quantizer to adjust the rate without retraining.

- We propose *data-driven* RAQ-VAE that generates a rate-adaptive codebook via a Seq2Seq model. This approach uses a single codebook and a training method, *cross-forcing*, to train recurrent networks to generate codebooks at different rates.

- Our experiments demonstrate that a single RAQ-VAE model achieves or even outperforms the performance of multiple VQ-VAE models trained at fixed rates, using the same encoder-decoder architecture.

## 2 BACKGROUND

**Vector-Quantized Autoencoder** VQ-VAEs (Van Den Oord et al., 2017; Razavi et al., 2019) can successfully represent meaningful features that span multiple dimensions of data space by discretizing continuous latent variables to the nearest code vector in its codebook. In VQ-VAE, learning of discrete representations is achieved by quantizing the encoded latent variables to their nearest neighbors in a trainable codebook and decoding the input data from the discrete latent variables. To represent the data $\mathbf{x} \in \mathbb{R}^{3 \times H \times W}$ from dataset $\mathcal{D}$ discretely, a codebook $\mathbf{e}$ consisting of $K$ learnable code vectors $\{e_i\}_{i=1}^{K} \subset \mathbb{R}^d$ is employed. The quantized discrete latent variable $\mathbf{z}_q(\mathbf{x}|\mathbf{e})$ is decoded to reconstruct the data $\mathbf{x}$. The quantizer $Q$ modeled as deterministic categorical posterior maps a continuous latent representation $f_\phi(\mathbf{x})$ of the data $\mathbf{x}$ by a deterministic encoder $f_\phi$ to $\mathbf{z}_q(\mathbf{x}|\mathbf{e})$ by finding the nearest neighbor in the $D$-dimensional codebook $\mathbf{e} = \{e_i\}_{i=1}^{K}$ as

$$\mathbf{z}_q(\mathbf{x}|\mathbf{e}) = Q\left(f_\phi(\mathbf{x})|\mathbf{e}\right) = \underset{e_i \in \{e_i\}_{i=1}^{K}}{\arg\min} \|f_\phi(\mathbf{x}) - e_i\|. \tag{1}$$

The quantized representation is fixed to $\log_2 K$ bits for the index $i$ of the selected code vector $e_i$ of the codebook of size $K$. The deterministic decoder $f_\theta$ reconstruct the data $\mathbf{x}$ from the quantized discrete latent variable $\mathbf{z}_q(\mathbf{x}|\mathbf{e})$ as $\hat{\mathbf{x}} = f_\theta\left(\mathbf{z}_q(\mathbf{x}|\mathbf{e})|\mathbf{e}\right)$. During the training process, the encoder $f_\phi$, decoder $f_\theta$, and codebook $\mathbf{e}$ are jointly optimized to minimize the loss $\mathcal{L}_{\text{VQ}}\left(\phi, \theta, \mathbf{e}; \mathbf{x}\right) =$

$$\underbrace{\log p_\theta(\mathbf{x}|\mathbf{z}_q(\mathbf{x}|\mathbf{e}))}_{\mathcal{L}_{\text{recon}}} + \underbrace{\left\|\text{sg}\left[f_\phi(\mathbf{x})\right] - \mathbf{z}_q(\mathbf{x}|\mathbf{e})\right\|_2^2}_{\mathcal{L}_{\text{embed}}} + \underbrace{\beta\left\|\text{sg}\left[\mathbf{z}_q(\mathbf{x}|\mathbf{e})\right] - f_\phi(\mathbf{x})\right\|_2^2}_{\mathcal{L}_{\text{commit}}} \tag{2}$$

where $\text{sg}[\cdot]$ is the *stop-gradient* operator. The $\mathcal{L}_{\text{recon}}$ is the reconstruction loss between the input data $\mathbf{x}$ and the reconstructed decoder output $\hat{\mathbf{x}}$. The two $\mathcal{L}_{\text{embed}}$ and $\mathcal{L}_{\text{commit}}$ apply only to codebook variables and encoder weight with a weighting hyperparameter $\beta$ to prevent fluctuations from

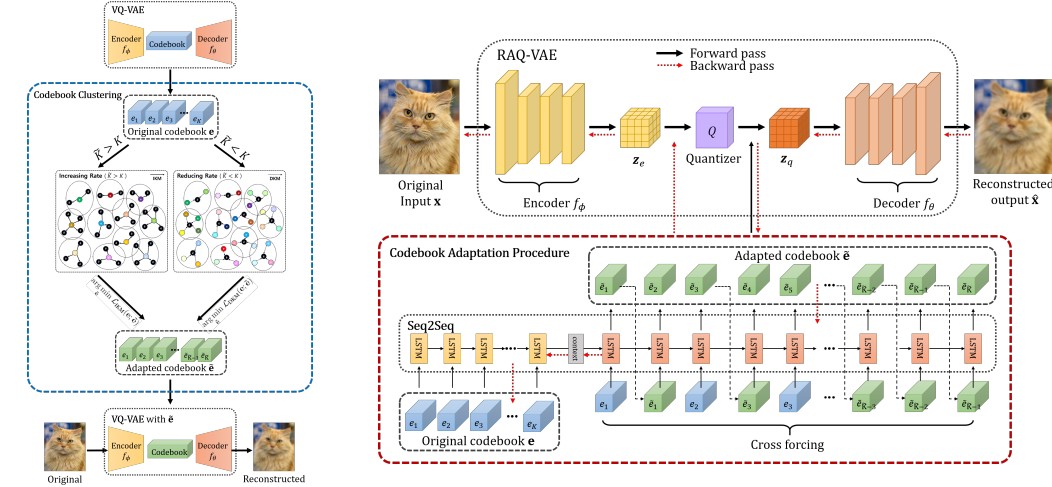

Figure 1: **Model-based RAQ-VAE** *(left)*: the model-based approach clusters the codebook **e** of a trained VQ-VAE model with separate tasks for reducing or increasing to the adapted codebook ẽ and applies it to the model. **Data-driven RAQ-VAE** *(right)*: the data-driven approach trains a Seq2Seq-based codebook adaptation procedure utilizing the baseline VQ-VAE model with data, where the gradient flow of the codebook passes through the Seq2Seq model.

one code vector to another. Since the quantization process is non-differentiable, the codebook loss is typically approximated via a straight-through gradient estimator (Bengio et al., 2013), such as $\partial \mathcal{L}/\partial f_\phi(\mathbf{x}) \approx \partial \mathcal{L}/\partial \mathbf{z}_q(\mathbf{x})$. Both conventional VAE (Kingma & Welling, 2013) and VQ-VAE (Van Den Oord et al., 2017) have objective functions consisting of the sum of reconstruction error and latent regularization. To improve performance and convergence rate, exponential moving average (EMA) update is usually applied for the codebook optimization (Van Den Oord et al., 2017; Razavi et al., 2019) (for more details in supplementary material A.1).

**Seq2Seq** The sequence-to-sequence (Seq2Seq) (Sutskever et al., 2014) model is widely used in sequence prediction tasks such as language modeling and machine translation (Dai & Le, 2015; Luong et al., 2016; Ranzato et al., 2016). The model employs an initial LSTM, called the encoder, to process the input sequence sequentially and produce a substantial fixed-dimensional vector representation, called the context vector. The output sequence is then derived by a further LSTM, the decoder. In particular, the decoder is conditioned on the input sequence, distinguishing it as a distinct component within the architecture. During training, the Seq2Seq model typically uses *teacher forcing* (Williams & Zipser, 1989), where the target sequence is provided to the decoder at each time step, instead of the decoder using its own previous output as input. This method helps the model converge faster by providing the correct context during training.

## 3 RATE-ADAPTIVE VQ-VAE

Although VQ-VAE has been successfully applied to various domains, it still faces scalability limitation. In particular, the common fixed-rate VQ-VAE model requires modifying the codebook size $K$ when processing different datasets (see (Razavi et al., 2019; Esser et al., 2021), using as many as 16384 and as few as 512 codebook sizes are used). Furthermore, adjusting the computational load requires retraining the model, which poses additional challenges. To overcome these limitations, we introduce two Rate-Adaptive VQ-VAE (RAQ-VAE) frameworks, which can adjust the rate of VQ-VAE through increasing or decreasing of the codebook size $K$. The outline of the RAQ-VAE framework is shown in Figure 1. RAQ-VAE builds upon a codebook mapping $\Psi : (\mathbb{R}^d)^{\times K} \longrightarrow (\mathbb{R}^d)^{\times \widetilde{K}}$ for any integer $\widetilde{K} \in \mathbb{N}$ that can be either lower, i.e., $\widetilde{K} < K$, or higher, i.e., $\widetilde{K} > K$, than the original codebook size $K$. We design the mapping in two ways: (i) model-based RAQ-VAE; (ii) data-driven RAQ-VAE. Model-based RAQ-VAE (Sec. 3.1) can obtain rate-adaptive VQ through differentiable $k$-means clustering (DKM) (Cho et al., 2021) algorithm without any additional pa-

rameters. In addition, data-driven RAQ-VAE (Sec. 3.2) is an offline-trained RAQ-VAE method that adopts the codebook generative Sequence-to-Sequence (Seq2Seq) (Sutskever et al., 2014) model.

## 3.1 MODEL-BASED RATE-ADAPTIVE VQ-VAE

Previous attempts (Łańcucki et al., 2020; Tjandra et al., 2019; Zheng & Vedaldi, 2023) have proposed enhancing codebook learning by periodically clustering the codebook during model training. In contrast, we propose a model-based rate-adaptive VQ-VAE that performs online codebook clustering after the model has been trained. By loading a VQ-VAE model trained with the original codebook $\mathbf{e}$ and dynamically clustering the codebook to the adapted codebook $\tilde{\mathbf{e}}$. This allows the vector quantizer to adapt to nuanced patterns within the overall model, providing flexibility and scalability (See Figure 1).

**Codebook Clustering**  To achieve the desired rate for the adapted codebook size $\widetilde{K}$ $(= |\tilde{\mathbf{e}}|)$, we derive the clustered codebook $\tilde{\mathbf{e}}$ from the original codebook $\mathbf{e}$. Details of the codebook clustering formulation are provided in supplementary material A.2. To ensure that the clustering process is effectively integrated into the trained VQ-VAE model, we employ a differentiable $k$-means clustering (DKM) algorithm (Cho et al., 2021). This algorithm, originally proposed for DNN model compression, uses an attention-based weight clustering method. We use the DKM algorithm for VQ codebook clustering, focusing on the fine-tuning of clustered codebooks and VQ-VAE model architectures. Additionally, we utilize DKM for codebook incrementation (inverse functionalization process) to handle scenarios requiring an increase in codebook size.

**Reducing the Rate** $(\widetilde{K} < K)$  In the rate reduction task, DKM can perform iterative differentiable codebook clustering on $\widetilde{K}$ clusters. Let $\mathbf{C}$ represent the cluster centers and vector $\mathbf{e}$ represent the original codebook. The DKM algorithm for VQ codebook operates as follows:

- Initialize a centroid $\mathbf{C} = \{c_j\}_{j=1}^{\widetilde{K}}$ either by randomly selected $\widetilde{K}$ codebook vectors from $\mathbf{e}$ or using $k$-means++. The last known $\mathbf{C}$ from the previous batch is used for all following iterations.

- Calculate the distance between the original codebook vector $e_i$ and initialized centroid $c_j$ using Euclidean distance as the distance metric $D_{i,j} = -f(e_i, c_j)$ with its matrix $\boldsymbol{D}$.

- To obtain the attention matrix $\boldsymbol{A}$, derive each row of $\boldsymbol{A}$ where $A_{i,j} = \frac{\exp\left(\frac{D_{i,j}}{\tau}\right)}{\sum_k \exp\left(\frac{D_{i,k}}{\tau}\right)}$ represents the attention from $e_i$ and $c_j$ with a softmax temperature $\tau$.

- Get a centroid candidate $\widetilde{\mathbf{C}} = \{\tilde{c}_j\}_{j=1}^{\widetilde{K}}$ by summing all the attentions for each centroid by computing $\tilde{c}_j = \frac{\sum_i A_{i,j} e_i}{\sum_i A_{i,j}}$ and update $\mathbf{C}$ with $\widetilde{\mathbf{C}}$.

- Repeat this process until $|\mathbf{C} - \widetilde{\mathbf{C}}| \leq \epsilon$ at which point DKM has converged or the iteration limit reached, then compute $\boldsymbol{A}\mathbf{C}$ to get $\tilde{\mathbf{e}}$.

The above iterative process can be summarized as follows:

$$\tilde{\mathbf{e}} = \underset{\tilde{\mathbf{e}}}{\arg\min}\, \mathcal{L}_{\text{DKM}}(\mathbf{e}; \tilde{\mathbf{e}}) = \underset{\mathbf{C}}{\arg\min} |\mathbf{C} - \boldsymbol{A}\mathbf{C}| = \underset{\mathbf{C}}{\arg\min} \sum_{j=1}^{\widetilde{K}} \left| c_j - \frac{\sum_i A_{i,j} e_i}{\sum_i A_{i,j}} \right| \tag{3}$$

In (Cho et al., 2021), the authors implemented DKM for soft-weighted cluster assignment and hardness can be enforced to provide weighted clustering constraints. In the softmax operation, the temperature $\tau$ can be used to control the level of hardness. At the end of the DKM process, we use the last attention matrix $\boldsymbol{A}$ to snap each codebook vector to the nearest centroid and finish clustering the codebook.

**Increasing the Rate** $(\widetilde{K} > K)$  While $k$-means clustering is effective for compressing code vectors, it has algorithmic limitations that prevent the augmentation of additional codebooks. To address this, we introduce the inverse functional DKM (IKM), a technique for increasing the number of

codebooks. This iterative method aims to approximate the posterior distribution of an existing generated codebook. We use maximum mean discrepancy (MMD) to compare the distribution difference between the base codebook and the clustered generated codebook, where MMD is a kernel-based statistical test technique that measures the similarity between two distributions (Gretton et al., 2012).

Assuming the original codebook vector $\mathbf{e}$ of size $K$ already trained in the baseline VQ-VAE, the process of generating the codebook $\tilde{\mathbf{e}}$ using the IKM algorithm is performed as follows:

- Initialize a $d$-dimensional adapted codebook vector $\tilde{\mathbf{e}} = \{\tilde{e}_i\}_{i=1}^{\widetilde{K}}$ as $\tilde{\mathbf{e}} \sim \mathcal{N}(0, d^{-\frac{1}{2}} \boldsymbol{I}_{\widetilde{K}})$
- Cluster $\tilde{\mathbf{e}}$ via the DKM process (equation 3): $g_{\text{DKM}}(\tilde{\mathbf{e}}) = \underset{g_{\text{DKM}}(\tilde{\mathbf{e}})}{\arg\min} \mathcal{L}_{\text{DKM}}(\tilde{\mathbf{e}}; g_{\text{DKM}}(\tilde{\mathbf{e}}))$.
- Calculate the MMD between the true original codebook $\mathbf{e}$ and the DKM clustered $g_{\text{DKM}}(\tilde{\mathbf{e}})$.
- Optimize $\tilde{\mathbf{e}}$ to minimize the MMD objective $\mathcal{L}_{\text{IKM}}(\mathbf{e}; \tilde{\mathbf{e}}) = \text{MMD}(\mathbf{e}, g_{\text{DKM}}(\tilde{\mathbf{e}})) + \lambda ||\tilde{\mathbf{e}}||^2$.

where $\lambda$ is the regularization parameter controlling the strength of the L2 regularization term. The IKM process can be summarized as $\tilde{\mathbf{e}} = \underset{\tilde{\mathbf{e}}}{\arg\min} \mathcal{L}_{\text{IKM}}(\mathbf{e}; \tilde{\mathbf{e}})$. Since DKM does not block gradient flow, we easily can update the codebook $\tilde{\mathbf{e}}$ using stochastic gradient descent (SGD) as $\tilde{\mathbf{e}} = \tilde{\mathbf{e}} - \eta \nabla \mathcal{L}_{\text{IKM}}(\mathbf{e}, \tilde{\mathbf{e}})$. With DKM and IKM, the generated codebook $\tilde{\mathbf{e}}$ can be used to quantize the encoded vector as $\mathbf{z}_q(\mathbf{x}|\tilde{\mathbf{e}})$ at different rates without adding any model parameters to the trained VQ-VAE. Since DKM does not block gradient flow, it is easy to change the codebook cluster assignments even during offline and online training. During offline training, the clusters that are best suited in terms of VQ task loss are adopted. Although we do not focus on using multi-codebook with DKM (aim to leverage rate-adaptive codebook after trained), a multi-codebook VQ-VAE model can be easily implemented by tuning $\widetilde{K}$ with DKM during training and hierarchically optimizing the multi-codebook clusters with the model.

## 3.2 DATA-DRIVEN RATE-ADAPTIVE VQ-VAE

Seq2Seq models (Sutskever et al., 2014) have been widely used in machine translation to handle variable output sequences, where the length of sentences can differ significantly between languages. Inspired by this, we propose a Seq2Seq-based approach to generate rate-adaptive codebooks within the VQ-VAE framework. This section introduces the data-driven RAQ-VAE, which integrates a learning vector quantization layer with Seq2Seq model.

**Overview**  As shown in Figure 1, data-driven RAQ-VAE is constructed with a deterministic encoder-decoder pair, a trainable original codebook $\mathbf{e}$, and Seq2Seq model. The adapted codebook $\tilde{\mathbf{e}}$ is generated by the Seq2Seq model from the original codebook $\mathbf{e}$. Data-driven RAQ-VAE hierarchically quantizes the continuous latent representation $f_\phi(x)$ of data $\mathbf{x}$ into $\mathbf{z}_q(\mathbf{x}|\mathbf{e})$ and $\mathbf{z}_q(\mathbf{x}|\tilde{\mathbf{e}})$ via $\mathbf{e}$ and $\tilde{\mathbf{e}}$, respectively. Building on the conventional VQ-VAE architecture, the data-driven RAQ-VAE learns the encoder-decoder pair while training the codebook $\mathbf{e}$ and its generative process $G_\psi$.

**Codebook Encoding**  The rate-adaptive codebook generation procedure, $G_\psi$, leverages LSTM cells in the Seq2Seq model to dynamically generate an adapted codebook $\tilde{\mathbf{e}}$ from the original codebook $\mathbf{e}$. The first step is to initialize the target codebook size $\widetilde{K}$. During training, the data-driven RAQ-VAE is trained with arbitrary codebook sizes $\widetilde{K}$. In the test phase, the Seq2Seq model generates the adapted codebook $\tilde{\mathbf{e}}$ at the desired rate specified by the user. This initialization sets the foundation for the encoding and decoding steps in Algorithm 1. Each vector of the original codebook $e_i$ is sequentially encoded by a set of LSTM cells. The hidden and cell states $(\boldsymbol{h}, \boldsymbol{c})$ capture the contextual information of each base codebook vector.

**Codebook Decoding via *cross-forcing***  The goal of Seq2Seq codebook generation is to reflect as much information as possible from the original codebook while generating a usable codebook for the VQ-VAE decoder. However, existing Seq2Seq training methods, such as teacher forcing (Williams & Zipser, 1989), may not be suitable when the target adapted codebook $\tilde{\mathbf{e}}$ consists of sequences that are much longer than the original codebook. Therefore, we propose *cross-forcing*, a hybrid approach combining teacher forcing and free running in professor forcing (Lamb et al., 2016). This is feasible because, unlike typical sequence prediction tasks, the order of the codebooks does not significantly

**Algorithm 1** Rate-adaptive codebook generation procedure $G_\psi$

**Input:** Original codebook $\mathbf{e} = \{e_i\}_{i=1}^K$
**Output:** Adapted codebook $\tilde{\mathbf{e}} = \{\tilde{e}_i\}_{i=1}^{\widetilde{K}}$
Initialize adapted codebook size $\widetilde{K}$,
hidden $\boldsymbol{h} = \{h_i\}_{i=1}^K$ and cell $\boldsymbol{c} = \{c_i\}_{i=1}^K$
▷ Codebook encoding
**for** $i = 1$ **to** $K$ **do**
    $h_i, c_i \leftarrow LSTM_\psi(e_i)$
**end for**
▷ Codebook decoding via *cross-forcing*
**for** $i = 1$ **to** $\widetilde{K}$ **do**
    **if** $i < 2K$ and $i$ is odd **then**
        $\tilde{e}_i \leftarrow LSTM_\psi(e_i, \boldsymbol{h}, \boldsymbol{c})$
    **else**
        $\tilde{e}_i \leftarrow LSTM_\psi(\tilde{e}_i, \boldsymbol{h}, \boldsymbol{c})$
    **end if**
**end for**
**Return:** $\tilde{\mathbf{e}} = G_\psi(\mathbf{e})$

**Algorithm 2** Training procedure of data-driven RAQ-VAE

**Input:** $\mathbf{x}$ (batch of training data)
**for** $\mathbf{x} \in$ train dataset $\mathcal{D}$ **do**
▷ Quantize encoder output $f_\phi(\mathbf{x})$ with $\mathbf{e}$.
    $\mathbf{z}_q(\mathbf{x}|\mathbf{e}) \leftarrow Q\left(f_\phi(\mathbf{x})|\mathbf{e}\right))$

▷ Generate $\tilde{\mathbf{e}}$ from Seq2Seq model $G_\psi$.
    $\tilde{\mathbf{e}} \leftarrow G_\psi(\mathbf{e})$ by Algorithm 1

▷ Quantize encoder output $f_\phi(\mathbf{x})$ with $\tilde{\mathbf{e}}$.
    $\mathbf{z}_q(\mathbf{x}|\tilde{\mathbf{e}}) \leftarrow Q\left(f_\phi(\mathbf{x})|\tilde{\mathbf{e}}\right))$

    $\hat{\mathbf{x}}_\mathbf{e}, \hat{\mathbf{x}}_{\tilde{\mathbf{e}}} \leftarrow f_\theta(\mathbf{z}_q(\mathbf{x}|\mathbf{e})), f_\theta(\mathbf{z}_q(\mathbf{x}|\tilde{\mathbf{e}}))$
    Compute $\mathcal{L}_{\text{VQ}}$ by equation 2.
    Compute $\mathcal{L}_{\text{RAQ}}$ by equation 4.
    $\phi, \theta, \mathbf{e} \leftarrow \text{Update}(\mathcal{L}_{\text{VQ}})$
    $\phi, \theta, \psi, \mathbf{e} \leftarrow \text{Update}(\mathcal{L}_{\text{RAQ}})$
**end for**
**Return:** $f_\phi, f_\theta, G_\psi, \mathbf{e}$

affect the outcome. In the decoding phase (as shown in Algorithm 1), teacher forcing is applied for odd steps that are less than twice the original codebook size ($2\widetilde{K}$), using the base code vector ($e_i$) as input. For even steps and beyond, *free running* (using the previous time step decoder output as input) is performed to dynamically train the VQ-VAE decoder with the generated codebook. The codebook decoding via cross-forcing is a key component of the data-driven approach. This technique helps ensure stable codebook generation at different rates. We have empirically validated its effectiveness in Appendix A.4.5.

**Training Procedure** To train the data-driven RAQ-VAE, we jointly optimize the base VQ-VAE and RAQ-VAE objectives to learn a good representation of the original codebook $\mathbf{e}$ and the rate-adaptive codebook generative process $G_\psi$. We formulate the constrained optimization $\mathcal{L}_{\text{RAQ}}$ to jointly update $G_\psi$ with $f_\phi$, $f_\theta$, and $\mathbf{e}$ as $\mathcal{L}_{\text{VQ}}\left(\phi, \theta, \mathbf{e}; \mathbf{x}\right) \geq \mathcal{L}_{\text{RAQ}}\left(\phi, \theta, \psi, \mathbf{e}; \mathbf{x}\right) =$

$$\log p_\theta\left(\mathbf{x}|\mathbf{z}_q(\mathbf{x}|G_\psi(\mathbf{e}))\right) + \left|\left|\text{sg}\left[f_\phi(\mathbf{x})\right] - \mathbf{z}_q(\mathbf{x}|G_\psi(\mathbf{e}))\right|\right|_2^2 + \beta\left|\left|\text{sg}\left[\mathbf{z}_q(\mathbf{x}|G_\psi(\mathbf{e}))\right] - f_\phi(\mathbf{x})\right|\right|_2^2. \quad (4)$$

where $\text{sg}[\cdot]$ is the *stop-gradient* operator. The data-driven RAQ-VAE jointly minimizes $\mathcal{L}_{\text{VQ}}$ (equation 2) and $\mathcal{L}_{\text{RAQ}}$ (equation 4). Back-propagating $\mathcal{L}_{\text{VQ}}$ induces the same gradient flows as the base VQ-VAE. Additionally, back-propagating $\mathcal{L}_{\text{RAQ}}$ induces a gradient flow to the Seq2Seq model, resulting in effective codebook generation. The overall training procedure for the proposed data-driven RAQ-VAE is summarized in Algorithm 2. During training, the Seq2Seq model dynamically generates codebooks and adapts to different rates at each training iteration.

# 4 RELATED WORK

**VQ-VAE and its Improvements** The VQ-VAE (Van Den Oord et al., 2017) has inspired numerous developments since its inception. Łańcucki et al. (2020); Williams et al. (2020); Zheng & Vedaldi (2023) proposed codeword reset and online clustering methods to address the problem of *codebook collapse* (Takida et al., 2022), thereby increasing the training efficiency of the codebook. Tjandra et al. (2019) introduced a conditional VQ-VAE that generates magnitude spectrograms for target speech using a multi-scale codebook-to-spectrogram inverter given the VQ-VAE codebook. SQ-VAE (Takida et al., 2022) incorporated stochastic quantization and a trainable posterior categorical distribution to enhance VQ-VAE performance, while Vuong et al. (2023) proposed VQ-WAE, based on SQ-VAE, using Wasserstein distance to ensure a uniform distribution of discrete representations. Several works have introduced substantial structural changes to VQ-VAE. Lee et al. (2022) proposed a two-step framework with Residual Quantized (RQ) VAE and RQ-Transform to generate high-resolution images using a single shared codebook. Mentzer et al. (2023) replaced VQ with Finite

Scalar Quantization (FSQ) to tackle codebook collapse. However, unlike previous works, we focus on achieving rate-adaptive VQ-VAE within a largely unchanged quantization scheme and VQ-VAE model architecture to improve its scalability for application not only to basic VQ-VAE models but also to its advanced models.

**Variable-Rate Neural Image Compression** Several studies have proposed variable-rate learning image compression frameworks based on different neural network architectures. Yang et al. (2020); Choi et al. (2019); Cui et al. (2020) introduced frameworks based on autoencoders, conditional autoencoders, and VAE structures, respectively. Variable-rate image compression has also been achieved in studies such as Song et al. (2021), which uses models based on the Spatial Feature Transform (SFT) for compression, and Johnston et al. (2018), which employs recurrent neural networks (RNNs) to achieve variable-rate compression by evaluating the distortion of individual patches to compute a weighted distortion. Duong et al. (2023) proposed learned transforms and entropy coding to enhance the linear transforms in existing codecs by systematizing the process into a single model that follows the rate-distortion curve. However, the integration of variable-rate image compression within the VQ-VAE framework remains an open question. Unlike these studies, our work focuses on embedding variable-rate compression directly into the VQ-VAE framework, maintaining the benefits of VQ while enhancing scalability and adaptability.

## 5 EXPERIMENTS

**Implementation** To demonstrate the advancement of the proposed RAQ-VAE, we adapt the conventional VQ-VAE (Van Den Oord et al., 2017) and the two-level hierarchical VQ-VAE (VQ-VAE-2) (Razavi et al., 2019) as baselines. We perform empirical evaluations on vision datasets: CIFAR10 ($32 \times 32$) (Krizhevsky et al.) and CelebA ($64 \times 64, 128 \times 128$) (Liu et al., 2015) for quantitative evaluation, and ImageNet ($256 \times 256$) (Russakovsky et al., 2015) for qualitative evaluation. We designed RAQ-VAE to adapt the conventional VQ-VAE and its improved model structures (Tjandra et al., 2019; Ott et al., 2019; Esser et al., 2021; Ramesh et al., 2021) to achieve multiple rates within a single model.

**Architecture** We use identical architecture and parameters for all methods, setting the default codeword (discrete latent) embedding dimension $d$ to 64 for CIFAR10 and CelebA, and to 128 for ImageNet. The codebook sizes range from 16 to 1024 for CIFAR10, 32 to 2048 for CelebA, and 128 to 4096 for ImageNet, with conventional VQ-VAE models trained on 'power of 2' sizes and RAQ-VAE models set to the middle of the range for both model-based and data-driven approaches. Details of the experimental settings are provided in supplementary material A.3.

**Evaluation Metrics** We quantitatively evaluated our method using peak-signal-to-noise-ratio (PSNR), structural similarity index measure (SSIM), reconstructed Fréchet inception distance (rFID) (Heusel et al., 2017), and codebook perplexity. PSNR measures the ratio between the maximum possible power of a signal and the power of the corrupted noise affecting data fidelity (Korhonen & You, 2012). SSIM assesses structural similarity between two images (Wang et al., 2004; Brunet et al., 2011). rFID assesses the quality of reconstructed images by comparing the distribution of features extracted from the test data with that of the original data. Codebook perplexity, defined as $e^{- \sum_i^{\widetilde{K}} p_{e_i} \log p_{e_i}}$ where $p_{e_i} = \frac{N_{e_i}}{\sum_j^{\widetilde{K}} N_{e_j}}$ and $N_{e_i}$ represents the encoded number for latent representation with codebook $e_i$, indicates a uniform prior distribution when the perplexity value reaches the codebook size ($\widetilde{K}$), meaning all codebooks are used equally.

### 5.1 MAIN RESULTS ON VISION TASKS

**Quantitative Evaluation** We empirically compare our RAQ-VAE models with the conventional VQ-VAE (Van Den Oord et al., 2017) for image reconstruction performance. We trained and evaluated each VQ-VAE with different codebook sizes ($K$) as a quantitative baseline, and then validated RAQ-VAE by adapting the rate (by adjusting $\widetilde{K}$) on a single model-based and data-driven RAQ-VAE model. Figure 2 shows the results, evaluated on the CIFAR10 and CelebA ($64 \times 64$) datasets. Under same compression rate and network architecture, all proposed RAQ-VAE models achieve

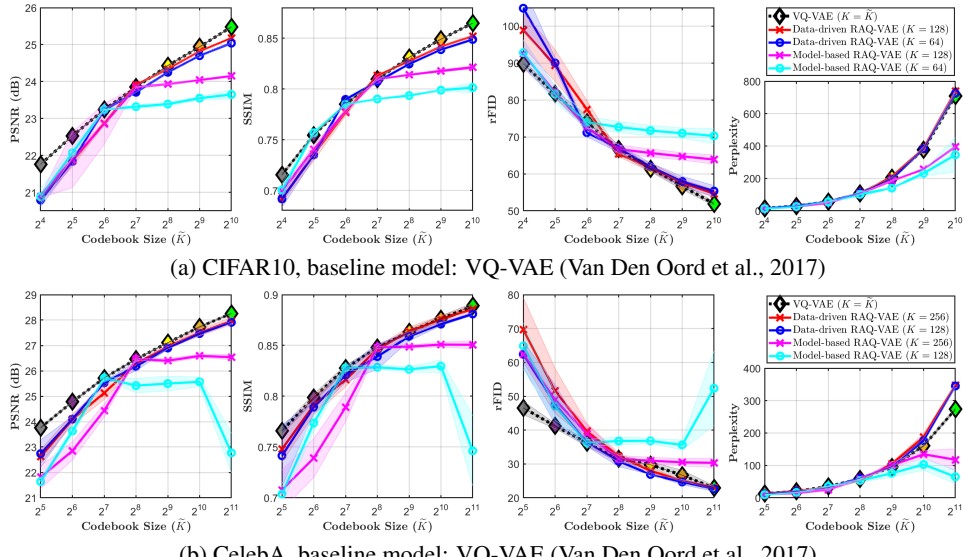

(a) CIFAR10, baseline model: VQ-VAE (Van Den Oord et al., 2017)

(b) CelebA, baseline model: VQ-VAE (Van Den Oord et al., 2017)

Figure 2: **Reconstruction performance** at different rates (adapted codebook sizes) evaluated on (a) CIFAR-10 and (b) CelebA. Higher values are better for PSNR, SSIM, and codebook perplexity, while lower values are better for rFID. The black lines represent separate VQ-VAE models trained individually for each codebook size $\widetilde{K}$. The colored lines represent a single RAQ-VAE model initially trained at a original codebook size $K$ and later adapted to different codebook sizes $\widetilde{K}$. The shaded area indicates the 95.45% confidence interval based on 4 runs with different training seeds.

performance close to that of multiple VQ-VAE models. When increasing the rate (codebook size), the data-driven RAQ-VAE achieves slightly lower results for the PSNR and SSIM metrics but better results in terms of rFID score, which evaluates perceptual image quality at the dataset level. In particular, the perplexity of the conventional VQ-VAE models shows low scores on CelebA, but the proposed data-driven RAQ-VAE performs better in terms of perplexity and rFID, especially at high bits per pixel (bpp). The model-based RAQ-VAE performs poorly overall, but in the task of reducing the rate, it achieves intermittently more reliable results on CIFAR10. Our proposed method is highly portable and reduces model complexity, considering the resources invested in each single VQ-VAE, since RAQ-VAE covers multiple fixed-rate VQ-VAE models with only a single model. (The model complexity of the baseline VQ-VAEs and our RAQ-VAEs are provided in A.3.3.)

**Qualitative Evaluation** For qualitative evaluation, we compare a single data-driven RAQ-VAE with VQ-VAEs trained at different rates (0.4375 bpp to 0.75 bpp) on ImageNet ($256 \times 256$). As seen in Figure 3, the VQ-VAEs (in the first row) are trained for each rate show that the quality decreases as the rate decreases, which is consistent with the results observed in the quantitative evaluation. When randomly selecting codebooks from a VQ-VAE model trained with $K = 4096$ (in the second row), we observe significant color changes, particularly at 0.5 bpp, where reconstructions retain structural similarity but show color distortions. However, the data-driven RAQ-VAE ($K = 512$), trained on a low-rate base codebook (0.5625 bpp), preserves the high-level semantic features and colors of the input image well with only a single model trained on the low-rate base codebook. Notably, it recovers fine details, like the cat's whiskers, far better than reconstructions using randomly selected codebooks. Although the image quality declines slightly at the lowest bpp, future work combining RAQ-VAE with advanced priors, such as PixelCNN (van den Oord et al., 2016) or PixelSNAIL (Chen et al., 2018), could further enhance the fidelity of generated images. Additional reconstructions can be found in the supplementary material A.4.4.

## 5.2 DETAILED ANALYSIS

**Codebook Usability** Following the observations of previous works (Wu & Flierl, 2020; Takida et al., 2022; Vuong et al., 2023), we note that as the codebook size increases, the codebook perplexity of data-driven RAQ-VAE also increases, leading to better reconstruction performance. In most VQ-

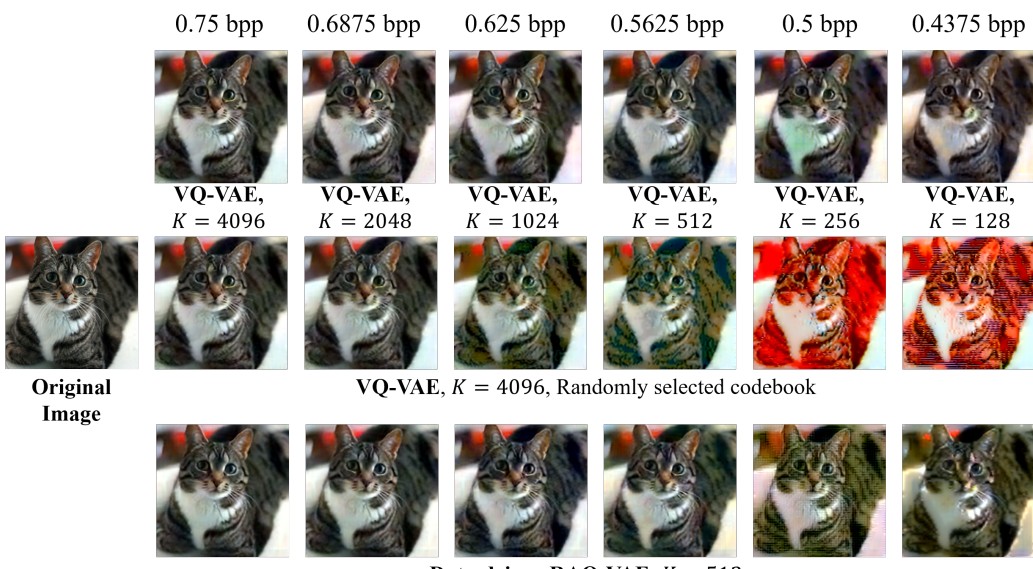

Figure 3: **Reconstructed samples** for the ImageNet dataset at different rates.

**rFID ↓**

| Method | Dataset | K | Adapted Codebook Size $\widetilde{K}$ | | | | | |
|---|---|---|---|---|---|---|---|---|
| | | | 2048 | 1024 | 512 | 256 | 128 | 64 |
| VQ-VAE-2 (Razavi et al., 2019) | CelebA | $\widetilde{K}$ | 10.11§ | 11.53 | 14.10 | 14.74§§ | 17.95 | 20.95 |
| VQ-VAE-2[†] (Razavi et al., 2019) | CelebA | 2048 | 10.11§ | 18.96 | 24.36 | 37.34 | 148.88 | 215.41 |
| **Model-based RAQ* (Ours)** | CelebA | 256 | 13.76 | 13.68 | **14.12** | **14.74**§§ | 20.29 | 53.46 |
| **Data-driven RAQ* (Ours)** | CelebA | 256 | **13.48** | **13.65** | 14.18 | 15.88 | **19.33** | **25.24** |
| | | | 512 | 256 | 128 | 64 | 32 | - |
| VQGAN (Esser et al., 2021) | ImageNet | $\widetilde{K}$ | 13.71§ | 14.38 | 18.96§§ | 22.48 | - | - |
| VQGAN[†] (Esser et al., 2021) | ImageNet | 512 | 13.71§ | 17.26 | 23.94 | 35.92 | 60.79 | - |
| **Model-based RAQ** (Ours)** | ImageNet | 128 | 18.81 | 18.89 | 18.96§§ | 24.58 | 34.16 | - |
| **Data-driven RAQ** (Ours)** | ImageNet | 128 | **15.91** | **17.54** | 19.83 | **22.55** | **31.54** | - |

Table 1: **Reconstruction performances** at different rates (according to $\widetilde{K}$) on CelebA ($128 \times 128$) test set and ImageNet ($256 \times 256$) validation set. † uses a single model for reconstructions with randomly selected codebooks. * denotes models trained with two-level hierarchical VQ-VAE (**VQ-VAE-2**) as in Razavi et al. (2019). ** denotes model trained with the **stage-1 VQGAN** as in Esser et al. (2021). § and §§ indicate results generated from the same model for the corresponding rates.

VAE frameworks, codebook perplexity is considered optimal when it approaches the codebook size, effectively utilizing the available resources when the codebook size is limited. As demonstrated in the main quantitative evaluation (see Figure 2), the data-driven RAQ-VAE outperforms conventional VQ-VAE in terms of codebook perplexity at higher bits per pixel (bpp). This improvement highlights the effectiveness of the Seq2Seq model in generating a codebook that the decoder can consistently and efficiently utilize. The ability of data-driven RAQ-VAE to maintain high codebook perplexity ensures better representation and reconstruction quality, proving its robustness in handling larger codebooks.

**Rate Adaptation** To demonstrate the rate adaptation performance, we validated RAQ-VAE by varying the adapted codebook size ($\widetilde{K}$). For the rate reduction task ($\widetilde{K} < K$), our experiments show that data-driven RAQ-VAE generally outperforms model-based RAQ-VAE in most aspects. However, on the CIFAR10, the model-based RAQ-VAE performs better at some rates. When a VQ-VAE model achieves high codebook perplexity, substantial performance can be achieved by simply clustering the codebook vectors (see more results in supplementary material A.4.1). For the rate

increasing task ($\widetilde{K} > K$), a more challenging adaptation task, data-driven RAQ-VAE successfully generated higher-rate codebooks, outperforming model-based RAQ-VAE and partially surpassing conventional VQ-VAE models trained at the same codebook size. This capability was especially pronounced on the CelebA dataset. For model-based RAQ-VAE, increasing the difference between the original and adjusted codebook sizes resulted in noticeable performance degradation, exposing the limitations of the current implementation. However, the model-based approach can be advantageous for practitioners with limited computing resources as it allows them to just load and apply codebook embeddings to the pre-trained VQ-VAE models without the need for additional training. Although performance limitations remain, we suggest that adapting rates via the model-based approach could be another promising direction for future research. It is likely that large models such as ViT-VQGAN (Yu et al., 2022) would experience even greater computational overhead compared to CNN-based models, making this approach potentially beneficial.

**Applicability**  To demonstrate the broader applicability of our methodology, we extend our approach to the two-level hierarchical VQ-VAE (VQ-VAE-2) model (Razavi et al., 2019) and the stage-1 VQGAN model (Esser et al., 2021) as the baseline models. The VQ-VAE-2 model is an extension of the original VQ-VAE framework by incorporating a hierarchical structure that allows for improved representation and reconstruction capabilities. The VQGAN enhances the encoding process in the first stage by incorporating adversarial and perceptual losses (Johnson et al., 2016; Zhang et al., 2018), allowing for the generation of images with finer details. Table 1 provides the reconstruction performance according to the adapted codebook size $\widetilde{K}$ for the baseline models. Although models trained at specific codebook sizes (first rows) achieve slightly better reconstruction, RAQ-VAE offers a flexible, efficient solution by covering multiple rates with a single adaptive model. In most cases, the data-driven RAQ method outperforms the model-based approach. As shown in Figure 3, training with a large codebook and then randomly selecting the codebook leads to significant degradation when more than half of the codebook is removed. Applying our rate-adaptive quantization to VQ-VAE-2 and VQGAN not only preserves the performance of hierarchical or GAN-based models but also provides the flexibility to adapt to different rates without retraining. This demonstrates that RAQ-VAE extends beyond VQ-VAE, offering a versatile solution for more advanced VQ-based models, with significant potential in data reconstruction and generation tasks.

## 6 CONCLUSION

We introduced the Rate-Adaptive VQ-VAE (RAQ-VAE) framework, which addresses the scalability limitations of conventional VQ-VAEs through two novel codebook representation methods. Our experiments demonstrate that single RAQ-VAE model achieves superior reconstruction performance across multiple rates without the need for retraining. The ability to dynamically adjust rates without retraining makes it particularly beneficial for resource-constrained environments, simplifying model deployment and management. This rate-adaptive capability provides significant flexibility for applications that require dynamic compression levels, such as variable-rate image and video compression (Xu et al., 2023) or real-time end-to-end communication systems (Park et al., 2020). Although performance limitations remain, future work could further enhance stability and performance, increasing the overall value of our framework. With its proven versatility, RAQ-VAE has the potential to drive significant advances in both the theoretical and practical fields of machine learning.

**Ethics Statement**  RAQ-VAE is designed as a rate-adaptive extension of VQ-VAE and can be applied in all domains where VQ-based methods are used. As with all generative models, attention should be given to potential biases in the training data, as these can affect generated outputs. RAQ-VAE does not introduce any new ethical concerns beyond those inherent in VQ-VAE models.

**Reproducibility Statement**  Appendix A.3 provides details of the experiments. The complete code necessary to reproduce our experiments is included in the supplementary material.

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

# A APPENDIX / SUPPLEMENTARY MATERIAL

## A.1 VQ-VAE CODEBOOK UPDATES WITH EXPONENTIAL MOVING AVERAGES (EMA)

At training step $t$, the $n_i$ encoder outputs $\{f_\phi(x_1), f_\phi(x_2), ..., f_\phi(x_{n_i})\}$ from codebook $e_i$ for the mini-batch data $\{x_1, x_2, ..., x_{n_i}\}$ are updated with count $N_i^{(t)}$ and mean value $m_i^{(t)}$ as follows:

$$N_i^{(t)} := \gamma \cdot N_i^{(t-1)} + (1 - \gamma) \cdot n_i^{(t)}$$

$$m_i^{(t)} := \gamma \cdot m_i^{(t-1)} + (1 - \gamma) \cdot \sum_{j}^{n_i^{(t)}} f_\phi(x_j)^{(t)} \qquad (5)$$

$$e_i^{(t)} := \frac{m_i^{(t)}}{N_i^{(t)}}$$

where a $\gamma$ is a decay factor with a value between 0 and 1 (the default value $\gamma = 0.99$ was used in all of our experiments). The count $N_i^{(t)}$ represents the encoder hidden states that have $e_i$ as it's nearest neighbor. $N_i^{(0)}$ is initially set as zero.

## A.2 CODEBOOK CLUSTERING OF MODEL-BASED RAQ-VAE

Given a set of the original codebook representations $\mathbf{e} = \{e_i\}_{i=1}^{K}$, we aim to partition the $K$ code vectors into $\widetilde{K}(\leq K)$ code vectors $\tilde{\mathbf{e}} = \{\tilde{e}_i\}_{i=1}^{\widetilde{K}}$. Each codebook vector resides in a $D$-dimensional Euclidean space. Using the codebook assignment function $g(\cdot)$, then $g(e_i) = j$ means $i$-th given codebook assigned $j$-th clustered codebook. Our objective for codebook clustering is to minimize the discrepancy $\mathcal{L}$ between the given codebook $\mathbf{e}$ and clustered codebook $\tilde{\mathbf{e}}$:

$$\arg\min_{\tilde{\mathbf{e}}, g} \mathcal{L}(\mathbf{e}; \tilde{\mathbf{e}}) = \arg\min_{\tilde{\mathbf{e}}, g} \sum_{i=1}^{\widetilde{K}} ||e_i - \tilde{e}_{g(e_i)}|| \qquad (6)$$

with necessary conditions

$$g(e_i) = \arg\min_{j \in 1, 2, ..., \widetilde{K}} ||e_i - \tilde{e}_j||, \quad \tilde{e}_j = \frac{\sum_{i:g(e_i)=j} e_i}{N_j} \qquad (7)$$

where $N_j$ is the number of samples assigned to the codebook $\tilde{e}_j$.

## A.3 EXPERIMENT DETAILS

### A.3.1 ARCHITECTURES AND HYPERPARAMETERS

The model architecture for this study is based on the conventional VQ-VAE framework outlined in the original VQ-VAE paper (Van Den Oord et al., 2017), and is implemented with reference to the VQ-VAE-2 (Razavi et al., 2019) implementation repositories [1][2][3]. We are using the ConvResNets from the repositories. These networks consist of convolutional layers, transpose convolutional layers and ResBlocks. Experiments were conducted on two different computer setups: a server with 4 RTX 4090 GPUs and a machine with 2 RTX 3090 GPUs. PyTorch (Paszke et al., 2019), PyTorch Lightning (Falcon, 2019), and the AdamW (Loshchilov & Hutter, 2019) optimizer were used for model implementation and training. Evaluation metrics such as the Structural Similarity Index (SSIM) and the Frechet Inception Distance (rFID) were computed using implementations of pytorch-msssim [4] and pytorch-fid [5], respectively. The detailed model parameters are shown in Table 2. RAQ-VAEs are constructed based on the described VQ-VAE parameters with additional consideration of each parameter.

---

[1] https://github.com/mattiasxu/VQVAE-2
[2] https://github.com/rosinality/vq-vae-2-pytorch
[3] https://github.com/EugenHotaj/pytorch-generative
[4] https://github.com/VainF/pytorch-msssim
[5] https://github.com/mseitzer/pytorch-fid

| Method | Parameter | CIFAR10 | CelebA | ImageNet |
|---|---|---|---|---|
| VQ-VAE (Van Den Oord et al., 2017) | Input size | $32\times32\times3$ | $64\times64\times3$ | $224\times224\times3$ |
| | Latent layers | $8\times8$ | $16\times16$ | $56\times56$ |
| | Hidden units | 128 | 128 | 256 |
| | Residual units | 64 | 64 | 128 |
| | # of ResBlock | 2 | 2 | 2 |
| | Original codebook size ($K$) | $2^4 \sim 2^{10}$ | $2^5 \sim 2^{11}$ | $2^7 \sim 2^{12}$ |
| | Codebook dimension ($D$) | 64 | 64 | 128 |
| | $\beta$ (Commit loss weight) | 0.25 | 0.25 | 0.25 |
| | Weight decay in EMA ($\gamma$) | 0.99 | 0.99 | 0.99 |
| | Batch size | 128 | 128 | 32 |
| | Optimizer | AdamW | AdamW | AdamW |
| | Learning rate | 0.0005 | 0.0005 | 0.0005 |
| | Max. training steps | 195K | 635.5K | 2500K |
| **Model-based RAQ-VAE** | Original codebook size ($K$) | 64, 128 | 128, 256 | 512 |
| | Adapted codebook size ($\widetilde{K}$) | $2^4 \sim 2^{10}$ | $2^5 \sim 2^{11}$ | $2^6 \sim 2^{12}$ |
| | Max. DKM iteration | 200 | 200 | 200 |
| | Max. IKM iteration | 5000 | 5000 | 5000 |
| | $\tau$ of softmax | 0.01 | 0.01 | 0.01 |
| **Data-driven RAQ-VAE** | Original codebook size ($K$) | 64, 128 | 128, 256 | 512 |
| | Adapted codebook size ($\widetilde{K}$) | $2^4 \sim 2^{10}$ | $2^5 \sim 2^{11}$ | $2^6 \sim 2^{12}$ |
| | Max. Codebook size | 1024 | 2048 | 4096 |
| | Min. Codebook size | 8 | 16 | 64 |
| | Input size (Seq2Seq) | 64 | 64 | 128 |
| | Hidden size (Seq2Seq) | 64 | 64 | 128 |
| | # of reccuruent layers (Seq2Seq) | 2 | 2 | 2 |

Table 2: Architecture and hyperparameters of baseline VQ-VAE model and its RAQ-VAE model (Model-based RAQ-VAE and Data-driven RAQ-VAE)

### A.3.2 DATASETS AND PREPROCESSING

For the **CIFAR10** dataset, the training set is preprocessed using a combination of random cropping and random horizontal flipping. Specifically, a random crop of size $32 \times 32$ with padding of 4 using the 'reflect' padding mode is applied, followed by a random horizontal flip. The validation and test sets are processed by converting the images to tensors without further augmentation. For the **CelebA** dataset, the training set is preprocessed with a series of transformations. The images are resized and center cropped to $64 \times 64$, normalized, and subjected to random horizontal flipping. A similar preprocessing is applied to the validation set, while the test set is processed without augmentation. For the **ImageNet** dataset, the training set is preprocessed with a series of transformations. The images are resized $256 \times 256$ and center cropped to $224 \times 224$, normalized, and subjected to random horizontal flipping. A similar preprocessing is applied to the validation set, while the test set is processed without augmentation. These datasets are loaded into PyTorch using the provided data modules, and the corresponding data loaders are configured with the specified batch sizes and learning rate for efficient training (described in Table 2. The datasets are used as input for training, validation, and testing of the VQ-VAE model.

### A.3.3 MODEL COMPLEXITY

To provide a comprehensive understanding of the model complexity for the different datasets used in our experiments, we detail the number of parameters in the Encoder, Decoder, Quantizer, and Seq2Seq components of the trained models in Table 3 and 4. The table summarizes the number of model parameters counts for the CIFAR10 and CelebA datasets.

Moreover, we show the training/inference time in Table 5 and 6. The training and inference times were measured for both model-based and data-driven RAQ-VAE methods. These results highlight the trade-offs between our two methods and their potential applications depending on resource availability and performance requirements. Although the results show that the data-driven method has a higher computational cost, we expect that the benefit of achieving the rate will provide better flexibility and performance in different scenarios.

| Method | Encoder | Decoder | # params Quantizer | Seq2Seq | Total |
|---|---|---|---|---|---|
| **VQ-VAE** ($K = 1024$) | 196.3K | 262K | 65.5 K | - | 525K |
| **VQ-VAE** ($K = 512$) | 196.3K | 262K | 32.8K | - | 492K |
| **VQ-VAE** ($K = 256$) | 196.3K | 262K | 16.4K | - | 476K |
| **VQ-VAE** ($K = 128$) | 196.3K | 262K | 8.2K | - | 468K |
| **VQ-VAE** ($K = 64$) | 196.3K | 262K | 4.1K | - | 463K |
| **VQ-VAE** ($K = 32$) | 196.3K | 262K | 2.0K | - | 461K |
| **VQ-VAE** ($K = 16$) | 196.3K | 262K | 1.0K | - | 460K |
| **VQ-VAE** ($K = 1024$) (randomly selected codebook) | 196.3K | 262K | 65.5 K | - | 525K |
| **Data-driven RAQ-VAE** ($K = 128$) | 196.3K | 262K | 8.2K | 263.7K | 732K |
| **Data-driven RAQ-VAE** ($K = 64$) | 196.3K | 262K | 4.1K | 263.7K | 728K |
| **Model-based RAQ-VAE** ($K = 128$) | 196.3K | 262K | 8.2K | - | 468K |
| **Model-based RAQ-VAE** ($K = 64$) | 196.3K | 262K | 4.1K | - | 463K |

Table 3: Number of parameters for training our models on CIFAR10 dataset.

| Method | Encoder | Decoder | # params Quantizer | Seq2Seq | Total |
|---|---|---|---|---|---|
| **VQ-VAE** ($K = 2048$) | 196.3K | 262K | 131K | - | 590K |
| **VQ-VAE** ($K = 1024$) | 196.3K | 262K | 65.5 K | - | 525K |
| **VQ-VAE** ($K = 512$) | 196.3K | 262K | 32.8K | - | 492K |
| **VQ-VAE** ($K = 256$) | 196.3K | 262K | 16.4K | - | 476K |
| **VQ-VAE** ($K = 128$) | 196.3K | 262K | 8.2K | - | 468K |
| **VQ-VAE** ($K = 64$) | 196.3K | 262K | 4.1K | - | 463K |
| **VQ-VAE** ($K = 32$) | 196.3K | 262K | 2.0K | - | 461K |
| **VQ-VAE** ($K = 2048$) (randomly selected codebook) | 196.3K | 262K | 131K | - | 590K |
| **Data-driven RAQ-VAE** ($K = 256$)) | 196.3K | 262K | 16.4K | 263.7K | 740K |
| **Data-driven RAQ-VAE** ($K = 128$) | 196.3K | 262K | 8.2K | 263.7K | 732K |
| **Model-based RAQ-VAE** ($K = 256$) | 196.3K | 262K | 16.4K | - | 476K |
| **Model-based RAQ-VAE** ($K = 128$) | 196.3K | 262K | 8.2K | - | 468K |

Table 4: Number of parameters for training our models on CelebA dataset.

| Method | $K$ | Training time per epoch (s) | # params |
|---|---|---|---|
| **VQ-VAE / Model-based RAQ-VAE** | 64 | $18.09 \pm 0.256$ | 463K |
| **VQ-VAE / Model-based RAQ-VAE** | 256 | $18.43 \pm 0.1$ | 476K |
| **VQ-VAE / Model-based RAQ-VAE** | 1024 | $21.64 \pm 0.11$ | 525K |
| **Data-driven RAQ-VAE** | 256 | $514.97 \pm 08.17$ | 740K |

Table 5: Training time per epoch on on CelebA train set using a Nvidia RTX 3090 GPU.

## A.4 ADDITIONAL EXPERIMENTS

### A.4.1 REDUCING THE RATE

As analyzed in Section 5.1, data-driven RAQ-VAE generally outperforms model-based RAQ-VAE, but some rate-reduction results on CIFAR10 show that model-based RAQ-VAE performs much more stably than in the codebook increasing task. This indicates that simply clustering codebook vectors, without additional neural models like Seq2Seq, can achieve remarkable performance.

| Method | $\widetilde{K}$ | Inference time per epoch (s) |
|---|---|---|
| **VQ-VAE** ($K = 64$) | - | $1.86 \pm 0.10$ |
| **VQ-VAE** ($K = 256$) | - | $1.91 \pm 0.12$ |
| **VQ-VAE** ($K = 1024$) | - | $1.86 \pm 0.09$ |
| **Model-based RAQ-VAE** ($K = 256$) | 64 | $1.98 \pm 0.09$ |
| **Data-driven RAQ-VAE** ($K = 256$) | 64 | $3.05 \pm 0.11$ |
| **Model-based RAQ-VAE** ($K = 256$) | 1024 | $70.91 \pm 11.82$ |
| **Data-driven RAQ-VAE** ($K = 256$) | 1024 | $33.21 \pm 0.27$ |

Table 6: Inference time per epoch on CelebA test set using a Nvidia RTX 3090 GPU.

| Method | $\widetilde{K}$ | **CIFAR10** ($K = 1024$) | | |
|---|---|---|---|---|
| | | PSNR ↑ | rFID ↓ | Perplexity ↑ |
| **VQ-VAE** (baseline model) | - | 25.48 | 51.90 | 708.60 |
| **VQ-VAE** (random select) | 512 | 24.35 | 63.67 | **289.29** |
| | 256 | 22.81 | 78.00 | 111.77 |
| | 128 | 20.87 | 93.57 | 48.87 |
| **Model-based RAQ-VAE** | 512 | **24.62** | **55.78** | 285.68 |
| | 256 | **23.81** | **62.53** | **134.54** |
| | 128 | **23.07** | **69.45** | **73.17** |

| Method | $\widetilde{K}$ | **CelebA** ($K = 2048$) | | |
|---|---|---|---|---|
| | | PSNR ↑ | rFID ↓ | Perplexity ↑ |
| **VQ-VAE** (baseline model) | - | 28.26 | 22.89 | 273.47 |
| **VQ-VAE** (random select) | 1024 | 24.02 | 38.92 | **103.50** |
| | 512 | 18.99 | 71.64 | 49.59 |
| | 256 | 23.54 | 115.12 | 27.86 |
| **Model-based RAQ-VAE** | 1024 | **26.40** | **31.37** | 102.36 |
| | 512 | **25.24** | **39.07** | **53.45** |
| | 256 | **24.36** | **45.54** | **32.86** |

Table 7: Reconstuction performances for **rate-reduction task** according to adapted codebook size $\widetilde{K}$. The distortion (PSNR), perceptual similarity (rFID), and codebook usability (perplexity) are evaluated using the test set on CIFAR-10 an CelebA. Higher values are better for PSNR,and perplexity, while lower values are better for rFID.

In Table 7, the performance via codebook clustering was evaluated with different original/adapted codebook sizes $K$: 1024 / $\widetilde{K}$: 512, 256, 128 on CIFAR10 and $K$: 2048 / $\widetilde{K}$: 1024, 512, 256, 128 on CelebA. The conventional VQ-VAE preserved as many codebooks in the original codebook as in the adapted codebook, while randomly codebook-selected VQ-VAE results remained meaningless. Model-based RAQ-VAE adopted this baseline VQ-VAE model and performed clustering on the adapted codebook. Model-based RAQ-VAE shows a substantial performance difference in terms of reconstructed image distortion and codebook usage compared to randomly codebook-selected VQ-VAE. Even when evaluating absolute performance, it is intuitive that online codebook representation via model-based RAQ-VAE provides some performance guarantees.

### A.4.2 INCREASING THE RATE

In our proposed RAQ-VAE scenario, increasing the codebook size beyond the base size is a more demanding and crucial task than reducing it. The crucial step in building data-driven RAQ-VAE is to achieve higher rates from a fixed model architecture and compression rate, ensuring usability. Therefore, the codebook increasing task was the main challenge. The Seq2Seq decoding algorithm based on cross-forcing is designed with this intention.

In Figure 2, the codebook generation performance was evaluated with different original/adapted codebook sizes $K$: 64, 128 / $\widetilde{K}$: 64, 128, 256, 512, 1024 on CIFAR10 and $K$: 128, 256 / $\widetilde{K}$: 128, 256, 512, 1024, 2048 on CelebA datasets. As discussed in Section 5.1, data-driven RAQ-VAE out-

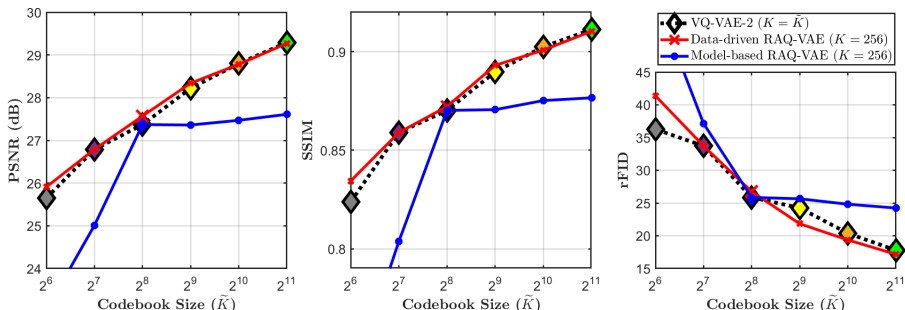

Figure 4: **Reconstruction performance** at different rates (adapted codebook sizes) evaluated on CelebA ($64 \times 64$) test set. In the graph, the black VQ-VAE-2s (Razavi et al., 2019) are separate models trained on each codebook size, while the RAQ-VAEs are one model per line.

performs model-based RAQ-VAE in the rate-increasing task and partially outperforms conventional VQ-VAE trained on the same codebook size ($K = \widetilde{K}$). This effect is particularly pronounced on CelebA.

However, increasing the difference between the original and adapted codebook sizes leads to a degradation of RAQ-VAE performance. This effect is more dramatic for model-based RAQ-VAE due to its algorithmic limitations, making its performance less stable at high rates. Improving the performance of model-based RAQ-VAE, such as modifying the initialization of the codebook vector, remains a limitation.

### A.4.3 ADDITIONAL QUANTITATIVE RESULTS

In Table 8 and 9, we present additional quantitative results for the reconstruction on CIFAR10 and CelebA datasets. The error indicates a 95.45% confidence interval based on 4 runs with different training seeds. Figure 4 shows the reconstruction performance using VQ-VAE-2 as the baseline model. The results demonstrate that the data-driven RAQ-VAE model significantly outperforms the original VQ-VAE-2 across multiple rates on the CelebA ($64 \times 64$)dataset.

### A.4.4 ADDITIONAL QUALITATIVE RESULTS

In Figure 5, we present additional qualitative results for the reconstruction on ImageNet dataset.

### A.4.5 EFFECTIVENESS OF *Cross-forcing*

We conducted additional experiments to demonstrate the effectiveness of the cross-forcing strategy, following a concern about the learning stability of this approach. In Table 10, the results compare the reconstruction performance of the data-driven RAQ-VAE ($K = 128$) **with** and **without** cross-forcing on the CelebA test dataset.

Our experiments demonstrate that the cross-forcing strategy is no less stable than the data-driven approach without cross-forcing when trained with the same four seeds. Furthermore, the performance improvement from cross-forcing becomes significant, particularly when operating with a codebook size equal to or larger than twice the original codebook size, as intended. This is in line with the general goal in machine learning of training models with a smaller original codebook size while still achieving better reconstruction performance at higher bitrates. It consistently improves performance metrics such as MSE, PSNR and rFID as the codebook size increases, making it an effective approach for tasks that demand higher bitrates without compromising model efficiency.

| Method | Bit Rate $\widetilde{K}$(bpp) | Codebook Usability Usage | Perplexity | Distortion PSNR | Perceptual Similarity rFID | SSIM |
|---|---|---|---|---|---|---|
| **VQ-VAE** $(K = \widetilde{K})$ | 1024 (0.625) | 972.66±2.97 | 708.60±7.04 | 25.48±0.02 | 51.90±0.51 | 0.8648±0.0005 |
| **VQ-VAE** $(K = \widetilde{K})$ | 512 (0.5625) | 507.52±0.51 | 377.08±5.92 | 24.94±0.01 | 56.65±0.91 | 0.8490±0.0003 |
| **VQ-VAE** $(K = \widetilde{K})$ | 256 (0.5) | 256±0 | 204.43±4.36 | 24.43±0.02 | 61.40±0.78 | 0.8310±0.0006 |
| **VQ-VAE** $(K = \widetilde{K})$ | 128 (0.4375) | 128±0 | 106.44±1.54 | 23.85±0.01 | 66.70±1.12 | 0.8096±0.0009 |
| **VQ-VAE** $(K = \widetilde{K})$ | 64 (0.375) | 64±0 | 55.64±0.27 | 23.24±0.01 | 74.00±1.64 | 0.7849±0.0009 |
| **VQ-VAE** $(K = \widetilde{K})$ | 32 (0.3125) | 32±0 | 29.25±0.13 | 22.53±0.02 | 81.68±1.01 | 0.7545±0.0009 |
| **VQ-VAE** $(K = \widetilde{K})$ | 16 (0.25) | 16±0 | 15.01±0.21 | 21.76±0.01 | 89.75±0.83 | 0.7156±0.0024 |
| **VQ-VAE** $(K = 1024)$ (random select) | 1024 (0.625) | 972.66±2.97 | 708.60±7.04 | 25.48±0.02 | 51.90±0.51 | 0.8648±0.0005 |
| | 512 (0.5625) | 498.38±1.85 | 289.29±16.67 | 24.35±0.11 | 63.67±2.49 | 0.8305±0.0056 |
| | 256 (0.5) | 253.01±0.66 | 111.77±21.53 | 22.81±0.38 | 78.00±5.07 | 0.7822±0.0100 |
| | 128 (0.4375) | 127.34±0.33 | 48.87±11.31 | 20.87±0.73 | 93.57±9.87 | 0.7254±0.0235 |
| | 64 (0.375) | 64±0 | 24.31±5.26 | 19.46±0.98 | 109.90±14.20 | 0.6720±0.0309 |
| | 32 (0.3125) | 32±0 | 13.50±1.45 | 17.76±1.12 | 126.57±15.89 | 0.6102±0.0350 |
| **Data-driven RAQ-VAE** $(K = 128)$ | 1024(0.625) | 971.21±4.14 | 724.91±15.34 | 24.85±0.02 | 57.03±1.34 | 0.8420±0.0008 |
| | 512 (0.5625) | 503.48±0.75 | 380.02±6.82 | 24.57±0.02 | 59.36±0.62 | 0.8326±0.0008 |
| | 256 (0.5) | 253.45±0.50 | 194.27±2.38 | 24.12±0.02 | 62.18±1.09 | 0.8193±0.0009 |
| | 128 (0.4375) | 128±0 | 109.65±3.50 | 23.71±0.01 | 66.89±1.07 | 0.8071±0.0014 |
| | 64 (0.375) | 64±0 | 55.64±0.27 | 23.08±0.02 | 71.84±0.31 | 0.7855±0.0005 |
| | 32 (0.3125) | 32±0 | 29.50±0.21 | 21.76±0.06 | 82.85±0.87 | 0.7384±0.0007 |
| | 16 (0.25) | 16±0 | 15.11±0.67 | 20.79±0.18 | 104.86±5.91 | 0.6918±0.0084 |
| **Model-based RAQ-VAE** $(K = 128)$ | 1024 (0.625) | 744.36±18.74 | 395.23±2.77 | 24.15±0.03 | 63.88±1..26 | 0.8213±0.0014 |
| | 512 (0.5625) | 430.06±11.58 | 256.23±7.50 | 24.04±0.03 | 64.74±0.96 | 0.8177±0.0012 |
| | 256 (0.5) | 244.61±3.13 | 185.02±3.31 | 23.93±0.01 | 65.65±1.12 | 0.8139±0.0010 |
| | 128 (0.4375) | 128±0 | 106.44±1.54 | 23.85±0.01 | 66.70±1.12 | 0.8096±0.0009 |
| | 64 (0.375) | 64±0 | 49.55±1.29 | 22.85±0.55 | 72.61±0.77 | 0.7780±0.0013 |
| | 32 (0.3125) | 32±0 | 25.65±0.76 | 21.88±0.75 | 82.12±1.74 | 0.7405±0.0046 |
| | 16 (0.25) | 16±0 | 13.79±0.06 | 20.89±0.04 | 95.03±0.34 | 0.6972±0.0010 |
| **Data-driven RAQ-VAE** $(K = 64)$ | 1024 (0.625) | 972.14±6.49 | 725.55±10.90 | 25.04±0.01 | 55.34±1.48 | 0.8487±0.0012 |
| | 512 (0.5625) | 506.38±1.23 | 382.43±10.58 | 24.70±0.02 | 57.91±1.42 | 0.8387±0.0011 |
| | 256 (0.5) | 255.52±0.48 | 196.17±9.95 | 24.25±0.02 | 61.96±1.00 | 0.8245±0.0012 |
| | 128 (0.4375) | 128±0 | 109.65±3.50 | 23.71±0.01 | 66.89±1.07 | 0.8071±0.0014 |
| | 64 (0.375) | 64±0 | 56.31±0.46 | 23.23±0.01 | 71.17±1.17 | 0.7897±0.0013 |
| | 32 (0.3125) | 32±0 | 29.62±0.66 | 21.84±0.09 | 90.04±1.44 | 0.7350±0.0038 |
| | 16 (0.25) | 16±0 | 15.11±0.67 | 20.79±0.18 | 104.86±5.91 | 0.6918±0.0084 |
| **Model-based RAQ-VAE** $(K = 64)$ | 1024 (0.625) | 706.20±115.18 | 345.50±107.06 | 23.65±0.13 | 70.30±2.02 | 0.8013±0.0051 |
| | 512 (0.5625) | 428.39±12.29 | 231.41±14.64 | 23.55±0.04 | 71.01±1.38 | 0.7988±0.0005 |
| | 256 (0.5) | 233.75±4.63 | 140.19±2.82 | 23.39±0.05 | 71.72±1.43 | 0.7935±0.0012 |
| | 128 (0.4375) | 125.07±1.58 | 101.16±16.04 | 23.32±0.05 | 72.68±1.47 | 0.7901±0.0008 |
| | 64 (0.375) | 64±0 | 55.64±0.27 | 23.24±0.01 | 74.00±1.64 | 0.7849±0.0009 |
| | 32 (0.3125) | 32±0 | 26.21±0.95 | 22.07±0.13 | 81.61±2.26 | 0.7569±0.0014 |
| | 16 (0.25) | 16±0 | 13.59±0.85 | 20.88±0.23 | 92.84±3.30 | 0.7004±0.0063 |

Table 8: **Reconstruction performance** on CIFAR10 dataset. The 95.45% confidence interval is provided based on 4 runs with different training seeds.

| Method | Bit Rate | Codebook Usability | | Distortion | Perceptual Similarity | |
|---|---|---|---|---|---|---|
| | $\widetilde{K}$ (bpp) | Usage | Perplexity | PSNR | rFID | SSIM |
| **VQ-VAE** ($K = \widetilde{K}$) | 2048 (0.6875) | 779.07±8.35 | 273.47±6.86 | 28.26±0.03 | 22.89±0.71 | 0.8890±0.0027 |
| **VQ-VAE** ($K = \widetilde{K}$) | 1024 (0.625) | 456.86±3.53 | 160.35±2.73 | 27.73±0.05 | 26.67±1.43 | 0.8763±0.0029 |
| **VQ-VAE** ($K = \widetilde{K}$) | 512 (0.5625) | 259.59±3.99 | 95.09±1.28 | 27.11±0.01 | 29.77±0.95 | 0.8636±0.0022 |
| **VQ-VAE** ($K = \widetilde{K}$) | 256 (0.5) | 144.44±2.49 | 57.86±0.91 | 26.46±0.03 | 31.53±1.01 | 0.8481±0.0009 |
| **VQ-VAE** ($K = \widetilde{K}$) | 128 (0.4375) | 80.26±0.99 | 34.98±0.39 | 25.72±0.04 | 36.25±0.98 | 0.8279±0.0027 |
| **VQ-VAE** ($K = \widetilde{K}$) | 64 (0.375) | 44.94±1.03 | 20.04±0.37 | 24.78±0.03 | 41.22±0.77 | 0.7986±0.0037 |
| **VQ-VAE** ($K = \widetilde{K}$) | 32 (0.3125) | 25.48±0.69 | 12.69±0.31 | 23.76±0.06 | 46.56±1.97 | 0.7660±0.0032 |
| **VQ-VAE** ($K = 2048$) (random select) | 2048 (0.625) | 779.07±8.35 | 273.47±6.86 | 28.26±0.03 | 22.89±0.71 | 0.8890±0.0027 |
| | 1024 (0.5625) | 384.31±6.76 | 103.50±3.28 | 24.02±1.10 | 38.92±3.27 | 0.7963±0.0201 |
| | 512 (0.5) | 210.69±9.23 | 49.59±4.54 | 18.99±1.40 | 71.64±8.27 | 0.7037±0.0221 |
| | 256 (0.4375) | 115.33±7.73 | 27.86±3.33 | 16.33±0.61 | 115.12±11.93 | 0.6353±0.0173 |
| **Data-driven RAQ-VAE** ($K = 256$) | 2048 (0.625) | 885.53±6.76 | 347.99±5.17 | 27.96±0.14 | 23.02±0.33 | 0.8858±0.0033 |
| | 1024 (0.5625) | 490.86±4.98 | 187.33±10.37 | 27.51±0.13 | 25.08±0.23 | 0.8758±0.0036 |
| | 512 (0.5) | 275.84±1.72 | 104.61±5.00 | 26.95±0.086 | 27.96±0.49 | 0.8637±0.0045 |
| | 256 (0.4375) | 144.79±1.21 | 52.63±0.28 | 26.29±0.054 | 32.34±0.86 | 0.8463±0.0030 |
| | 128 (0.375) | 80.21±4.27 | 32.23±3.87 | 25.13±0.26 | 39.67±2.29 | 0.8162±0.0071 |
| | 64 (0.3125) | 42.93±1.61 | 20.85±1.22 | 24.09±0.21 | 51.57±6.66 | 0.7912±0.0094 |
| | 32 (0.25) | 22.76±1.57 | 12.32±0.91 | 22.62±0.27 | 69.65±9.49 | 0.7479±0.0129 |
| **Model-based RAQ-VAE** ($K = 256$) | 2048 (0.625) | 704.17±108.04 | 117.53±33.57 | 26.54±0.10 | 30.34±1.39 | 0.8507±0.0041 |
| | 1024 (0.5625) | 460.77±26.98 | 134.48±11.26 | 26.59±0.06 | 30.49±1.10 | 0.8509±0.0021 |
| | 512 (0.5) | 279.53±9.48 | 100.64±08.94 | 26.40±0.08 | 30.95±0.98 | 0.8488±0.0017 |
| | 256 (0.4375) | 144.44±2.49 | 57.86±0.91 | 26.46±0.03 | 31.53±1.01 | 0.8481±0.0009 |
| | 128 (0.375) | 75.31±3.09 | 25.05±1.95 | 24.44±0.25 | 38.95±2.91 | 0.7890±0.0141 |
| | 64 (0.3125) | 41.66±1.22 | 14.73±0.56 3 | 22.85±0.36 | 48.96±1.13 | 0.7391±0.0192 |
| | 32 (0.25) | 22.96±0.90 | 10.16±0.95 | 21.81±0.45 | 62.46±0.00 | 0.7077±0.0195 |
| **Data-driven RAQ-VAE** ($K = 128$) | 2048 (0.625) | 891.13±7.11 | 345.25±5.15 | 27.91±0.04 | 22.64±0.76 | 0.8810±0.0013 |
| | 1024 (0.5625) | 490.15±14.39 | 176.71±6.19 | 27.47±0.07 | 24.67±0.80 | 0.8710±0.0016 |
| | 512 (0.5) | 272.60±2.08 | 96.87±2.68 | 26.90±0.05 | 26.90±0.04 | 0.8589±0.0044 |
| | 256 (0.4375) | 152.65±2.45 | 60.90±2.18 | 26.18±0.18 | 30.81±1.59 | 0.8391±0.0125 |
| | 128 (0.375) | 79.17±0.93 | 31.36±0.77 | 25.53±0.06 | 36.30±1.12 | 0.8209±0.0072 |
| | 64 (0.3125) | 42.71±1.66 | 19.78±2.31 | 24.10±0.11 | 47.63±5.82 | 0.7892±0.0067 |
| | 32 (0.25) | 22.42±1.92 | 11.43±2.14 | 22.74±0.54 | 62.39±3.76 | 0.7414±0.0304 |
| **Model-based RAQ-VAE** ($K = 128$) | 2048 (0.625) | 350.02±100.57 | 64.87±21.22 | 22.77±0.78 | 52.37±10.94 | 0.7463±0.0347 |
| | 1024 (0.5625) | 432.15±45.80 | 102.79±17.34 | 25.57±0.19 | 35.62±1.46 | 0.8296±0.0062 |
| | 512 (0.5) | 262.78±29.47 | 75.63±12.04 | 25.50±0.29 | 36.82±0.73 | 0.8265±0.0026 |
| | 256 (0.4375) | 153.16±5.46 | 53.22±4.62 | 25.42±0.28 | 36.78±1.27 | 0.8285±0.0022 |
| | 128 (0.375) | 80.26±0.99 | 34.98±0.39 | 25.72±0.04 | 36.25±0.98 | 0.8279±0.0027 |
| | 64 (0.3125) | 41.88±0.72 | 16.70±0.43 | 23.63±0.16 | 47.09±4.09 | 0.7736±0.0080 |
| | 32 (0.25) | 23.31±0.89 | 9.56±0.77 | 21.64±0.13 | 64.85±6.92 | 0.7037±0.0102 |

Table 9: **Reconstruction performance** on CelebA dataset. The 95.45% confidence interval is provided based on 4 runs with different training seeds.

| Method | $\widetilde{K}$ | MSE ↓ | PSNR ↑ | rFID ↓ | SSIM ↑ |
|---|---|---|---|---|---|
| **w/ cross-forcing** | 2048 (↑) | **1.618**±**0.016** | **27.91**±**0.04** | **22.64**±**0.76** | **0.8810**±**0.0013** |
| | 1024 (↑) | **1.794**±**0.027** | **27.47**±**0.07** | **24.67**±**0.80** | **0.8710**±**0.0016** |
| | 512 (↑) | **2.042**±**0.021** | **26.90**±**0.05** | **26.90**±**0.04** | **0.8589**±**0.0044** |
| | 256 (↑) | **2.412**±**0.101** | **26.18**±**0.18** | **30.81**±**1.59** | 0.8391±0.0125 |
| | 128 (-) | 2.801±0.039 | 25.53±0.06 | 36.30±1.12 | 0.8209±0.0072 |
| | 64 (↓) | 3.895±0.095 | 24.10±0.11 | 47.63±5.82 | 0.7892±0.0067 |
| | 32 (↓) | **5.357**±**0.630** | 22.74±0.54 | **62.39**±**3.76** | **0.7414**±**0.0304** |
| **w/o cross-forcing** | 2048 (↑) | 1.661±0.056 | 27.80±0.14 | 23.58±0.26 | 0.8789±0.0030 |
| | 1024 (↑) | 1.815±0.050 | 27.42±0.12 | 25.46±0.26 | 0.8705±0.0024 |
| | 512 (↑) | 2.068±0.059 | 26.85±0.12 | 27.81±0.42 | 0.8567±0.0046 |
| | 256 (↑) | 2.449±0.052 | 26.12±0.09 | 32.32±1.20 | **0.8407**±**0.0031** |
| | 128 (-) | **2.779**±**0.015** | 25.57±0.02 | **36.08**±**0.98** | **0.8261**±**0.0019** |
| | 64 (↓) | 3.860±0.237 | 24.15±0.26 | 45.13±2.79 | 0.7942±0.0154 |
| | 32 (↓) | 6.289±0.709 | 22.04±0.47 | 72.85±16.69 | 0.7338±0.0225 |

Table 10: Reconstruction performance of data-driven RAQ-VAE ($K = 128$) **with** or **without** cross-forcing on the CelebA test dataset

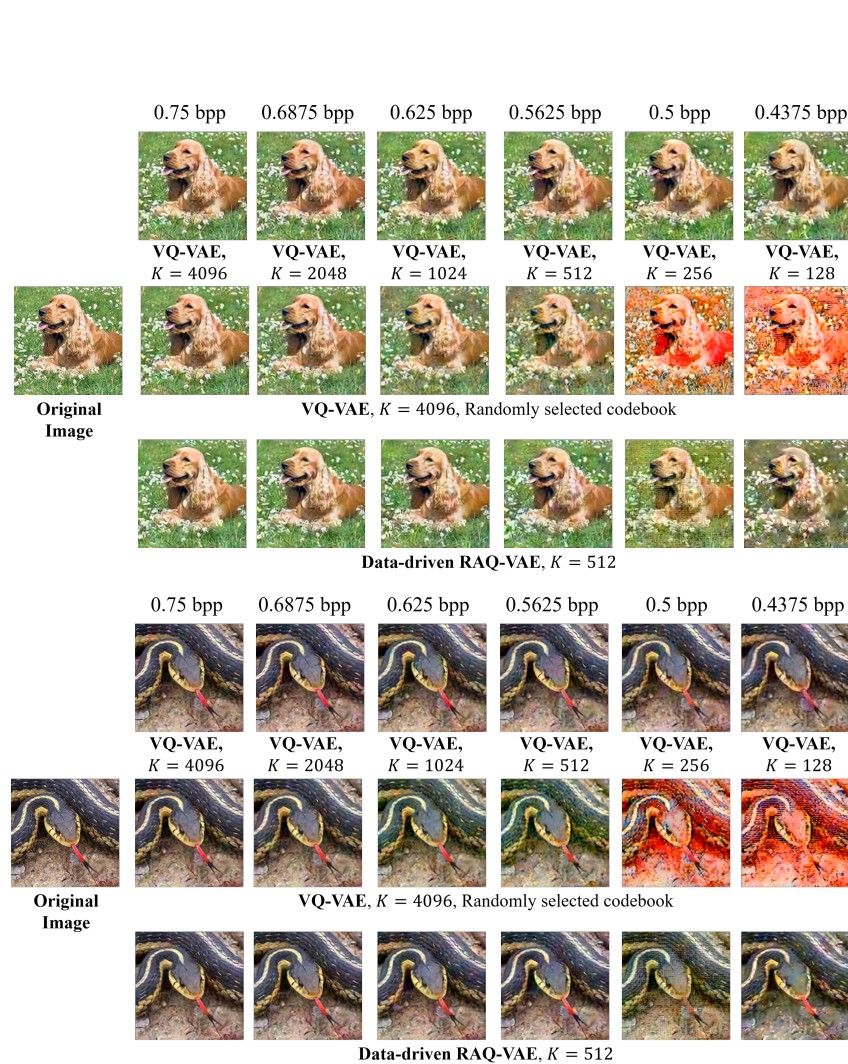

Figure 5: **Reconstructed images** for ImageNet dataset at different rates.

