# OpenReview forum: "RAQ-VAE: Rate-Adaptive Vector-Quantized Variational Autoencoder"
_ICLR.cc/2025/Conference — ICLR 2025 Conference Withdrawn Submission_

### Official Review · Reviewer_Cb5h · 2024-10-22

**Soundness:** 3
**Presentation:** 3
**Contribution:** 3
**Rating:** 6
**Confidence:** 4

**Summary:**

This paper focuses on changing the codebook of a VQ-VAE to have more (or fewer) codewords. It proposes two different approaches:

1. model-based: Train a VQ-VAE and then use differentiable k-means clustering to split (or consolidate) codewords and fine-tune the pre-trained VQ-VAE architecture.

2. data-based: Use a LSTM during VQ-VAE training to adapt the codebook to different sizes (i.e. train a VQ-VAE with 1024 codes, and use the LSTM during training to learn how to adapt the existing codewords to other sizes like 512, 8192, etc.).

The paper compares these two methods with training a VQ-VAE from scratch using other codebook sizes as well as randomly selecting codes from a pre-trained VQ-VAE with a large number of codewords.

**Strengths:**

**Problem:** This paper tackles an interesting problem: how to adapt the codebook of a pre-trained VQ-VAE to other applications where a different information rate is preferable.

**Writing:** The paper is well-written and easy to follow.

**Uniqueness / Novelty:** It is my view this application (or problem setting) is quite novel; I have not seen other work in the VQ-VAE literature addressing this question.

**Method:** Two different methods are proposed and explored (data-based and model-based). While both methods use existing machine learning components (i.e. this paper is not proposing a new building block or algorithm), they're used in a new setting that seems very fitting. A caveat to the last sentence is that the IKM algorithm to increase the codebook size seems to be a new contribution.

**Evaluation:** The empirical evaluation is comprehensive and sufficient to show both methods work (to varying degrees).

**Weaknesses:**

**Applicability:** It is unclear to me when the approaches presented in this work would be used. As likelihood-based or score-based generative models are often quite a bit more complex (and expensive) to train than VQ-VAEs, simply training a new VQ-VAE with the desired number of codebook vectors seems like it would not be an issue. Perhaps the authors can augment the introduction with specific examples where their approach would be desirable?

**Weak Baselines:** This work primarily compares with randomly selecting codes from a VQ-VAE to form the adapted codebook size. Unfortunately, many VQ-VAEs have low codebook utilization, so randomly selecting codes would likely incorporate codes that are never actually used by the model. A stronger (and very simple) baseline would be to select the $k$-most-used codes as measured on the validation (or training) set.

**Questions:**

See weaknesses. Additionally:

1. **Codebook utilization:** It is unclear how codebook utilization (or rate) changes with the adapted codebook sizes. Could this value be included for the experiments in Table 1 for the reconstructions? Specifically, for each batch of examples, count the number of unique codewords that are used and divide by the size of the codebook.

2. In Figure 2, it appears that the VQ-VAE trained with a codebook of size 1024 has the same (or lower) perplexity as both RAQ-VAE variants with K = 128 in both the top and bottom rows. How is the is the case when the VQ-VAE has such a larger vocab size? Is the codebook utilization for both of these models really around 10%?

---

> ### Author Response · Authors · 2024-11-13
>
> Thank you for your time reviewing our work and for your constructive feedback. We would like to address your questions and concerns below.
>
>
> > **Applicability:**  Perhaps the authors can augment the introduction with specific examples where their approach would be desirable?
>
> Recent studies, such as ViT-VQGAN, VQ-Diffusion, and Muse [B] have demonstrated the potential of vector-quantized image modelling by combining powerful transformers with VQGAN to generate high-quality images. However, these approaches involve significant computational overhead. For example, the stage 2 transformer reported in ViT-VQGAN has 1.4B parameters, and ViT-VQGAN has 1.7B parameters. VQ-Diffusion includes 370M parameters with its discrete diffusion process. Our RAQ-VAE framework is more applicable to these models with extensive computational resources, allowing for rate adaptation while efficiently leveraging these resources. While it may provide slightly better reconstruction quality for specific target rates compared to individually trained VQ-VAE models, the RAQ-VAE approach offers a highly flexible and computationally efficient alternative. For practitioners who prioritize scalability and efficiency in resource-poor environments, the ability of RAQ-VAE to dynamically adjust the rate without retraining multiple models is a significant advantage. Additionally, the RAQ-VAE framework reduces the overall complexity of model maintenance and deployment by eliminating the need to manage multiple models for different target rates. This simplification can streamline workflows and reduce operational costs. To strengthen our argument, we have shared the results of our method applied to a VQGAN with 72M parameters and evaluated it using rFID as a baseline.
>
> [B] Chang, Huiwen, et al. "Muse: Text-to-image generation via masked generative transformers." *ICML*. 2023.
>
> > **Weak Baselines:** Unfortunately, many VQ-VAEs have low codebook utilization, so randomly selecting codes would likely incorporate codes that are never actually used by the model.
>
> Our main goal was to demonstrate that our method achieves performance close to models trained from scratch with the corresponding codebook sizes. We agree that selecting the most-used codes based on their frequency in the validation or training set would serve as a stronger and more meaningful baseline compared to random selection. This approach is simple but more effective as it avoids the inclusion of rarely or never used codewords, which could bias the results due to low codebook utilization in VQ-VAEs. Due to an oversight by the main author in not properly listing co-authors, we had to withdraw the paper. However, we are grateful for your valuable feedback and will incorporate this improved baseline in our experiments for the next submission. Including this comparison will strengthen our evaluation and provide a clearer demonstration of the advantages of our method.
>
> > Codebook utilization: Could this value be included for the experiments in Table 1 for the reconstructions?
>
> We included experimental results on codebook usage and perplexity in Appendix Tables 8 and 9, but we omitted them from the main text because the trends were similar to those observed in the reconstruction experiments. For the baseline models corresponding to Table 1, such as VQ-VAE-2, the results showed similar trends as in Appendix Table 9. In the case of VQ-GAN, due to the relatively smaller codebook size, codebook usage in most experiments was close to 100%. In our next submission, we will include these values in Table 1 to provide a more comprehensive understanding of the model's behavior.
>
> > In Figure 2, it appears that the VQ-VAE trained with a codebook of size 1024 has the same (or lower) perplexity as both RAQ-VAE variants with K = 128 in both the top and bottom rows. How is the is the case when the VQ-VAE has such a larger vocab size? Is the codebook utilization for both of these models really around 10%?
>
> When using VQ-VAE as the baseline model with larger codebook sizes, such as $K=1024$ or $K=2048$, the perplexity (reflecting the average number of codewords used per image) tends to be around 10–20% of the total codebook size. However, if we consider codebook usage over the entire test set by counting codewords that are used at least once, we find higher utilization rates.
>
> For example, on the CelebA dataset with K=2048, the baseline VQ-VAE uses about 779 unique codewords across the test set, resulting in approximately 38.6% utilization. Our RAQ-VAE model uses around 891 unique codewords, achieving about 44% utilization. This indicates that while individual images may use a small subset of the codebook, collectively, a significant portion of the codebook is utilized across the dataset. We acknowledge that this can be counterintuitive, and we will provide a more detailed explanation and include these statistics in the revised manuscript.
>
> Thank you again for your valuable time and consideration.

---

### Official Review · Reviewer_Z2hp · 2024-11-01

**Soundness:** 3
**Presentation:** 3
**Contribution:** 3
**Rating:** 6
**Confidence:** 4

**Summary:**

This paper introduces two codebook representation methods for VQ-VAEs: model-based and data-driven Rate-Adaptive VQ-VAEs (RAQ-VAEs), which allow for changes in compression rates without the need to retrain the models. The model-based RAQ-VAE employs differentiable k-means clustering (DKM) and a newly proposed inverse functional DKM. The data-driven RAQ-VAE utilizes a Seq2Seq model to generate rate-adaptive codebooks. The two proposed methods provide trade-offs between performance and cost. Extensive experiments demonstrate the effectiveness of the proposed methods in achieving rate adaptation without retraining.

**Strengths:**

* The problem this paper addresses is clearly articulated, and the motivation for this work is clear.
* This paper proposes two complementary and intriguing algorithms that appropriately address rate adaptation without the need for retraining.
* Extensive experiments have been conducted to verify the capabilities of the proposed methods.
* Discussion about performance limitation is provided.

**Weaknesses:**

* Although some discussion regarding performance limitations is provided, the degradation of reconstruction quality is not negligible in certain cases. In particular, as shown in Figure 2, the model-based RAQ-VAE fails to improve reconstruction performance even when the codebook size is increased exponentially.

* There are several parts that could benefit from clearer explanations:
  - A proof of the inequality, \$ \mathcal{L}_\mathrm{VQ}(\phi, \theta, \mathbf{e}; \mathbf{x}) \geq \mathcal{L} _\mathrm{RAQ}(\phi, \theta, \psi, \mathbf{e}; \mathbf{x}) \$, in the Training Procedure on page 6 is not provided.
  - It appears that a sampling scheme for \$ \tilde{K} \$ is needed during the training of data-driven RAQ-VAEs, but I could not find an explanation for this.
  - The paper mentions that while image quality declines slightly at the lowest bits per pixel, future work combining RAQ-VAE with advanced priors could further enhance the fidelity of generated images, as stated in Section 5.1. However, the reconstruction performance of the autoencoder may serve as an upper bound for the performance of generative models based on that autoencoder. It seems that efforts to improve advanced priors are less likely to enhance the fidelity of generated images when starting from a poor autoencoder. A further discussion on how to overcome this bottleneck would strengthen the paper's arguments.

**Questions:**

* Could you please demonstrate a proof of the inequality \$ \mathcal{L} _\mathrm{VQ} \geq \mathcal{L} _\mathrm{RAQ} \$?
* Could you clarify how to sample \$ \tilde{K} \$ during the training of data-driven RAQ-VAEs?
* Could you provide additional insights on how to address the reconstruction performance bottleneck to enhance the performance of generative models?
* Minor comment: there are unpaired parentheses in Algorithm 2, e.g., $Q(f_\phi(x)|e)) $

---

> ### Author Response · Authors · 2024-11-13
>
> Thank you for your time reviewing and providing detailed feedback on our work. We would like to address your questions and concerns below.
>
>
> > Although some discussion regarding performance limitations is provided, the degradation of reconstruction quality is not negligible in certain cases. In particular, as shown in Figure 2, the model-based RAQ-VAE fails to improve reconstruction performance even when the codebook size is increased exponentially.
>
> We agree with your assessment. At the current level, the model-based approach is most effective when reducing the codebook size or making slight increases. The reduction methodology performs comparably to the data-driven approach. However, significantly increasing the codebook size using the model-based method does not yield substantial improvements in reconstruction performance. We acknowledge that further research and innovative methods are needed to address this limitation. We will include this discussion in the revised paper to clarify the optimal use cases and limitations of the model-based approach.
>
> > Could you please demonstrate a proof of the inequality? $L_{\text{VQ}} \ge L_{\text{RAQ}}$
>
> Thank you for bringing this to our attention. We realize that we currently lack a comprehensive definition of the constraints required to rigorously prove this inequality. To address this, we will provide a detailed explanation and define $L_{\text{RAQ}}$ and $L_{\text{VQ}}$ separately in the revised paper. This will clarify the relationship between the two loss functions and ensure that the inequality is properly justified.
>
> > It appears that a sampling scheme for $\tilde{K}$ is needed during the training of data-driven RAQ-VAEs, but I could not find an explanation for this. Could you clarify how to sample $\tilde{K}$ during the training of data-driven RAQ-VAEs?
>
> You raise a valid point. Improving priors may have limited impact on the fidelity of generated images when starting from an autoencoder with suboptimal reconstruction performance. While prior work has demonstrated that enhancing priors can improve image fidelity, the baseline model's capabilities ultimately serve as an upper bound. To overcome this bottleneck, one approach is to select a better-performing baseline model for the autoencoder. Alternatively, reducing the original codebook size in the baseline model can improve fidelity at lower bits per pixel (bpp), though this may introduce trade-offs in performance at higher bpp. We will include this discussion in the revised paper to provide additional insights on addressing the reconstruction performance bottleneck.
>
> > Minor comment: there are unpaired parentheses in Algorithm 2.
>
> Thank you for pointing out this error. We will correct the unpaired parentheses in Algorithm 2 in the revised version of the paper.
>
> Once again, we sincerely appreciate your thoughtful feedback. Your comments have been invaluable in helping us improve our work, and we will incorporate the suggested revisions to strengthen our manuscript.

---

### Official Review · Reviewer_GXkP · 2024-11-03

**Soundness:** 2
**Presentation:** 2
**Contribution:** 2
**Rating:** 5
**Confidence:** 4

**Summary:**

This paper extends the VQ-VAE to the rate-adaptive VQ-VAE (RVQ-VAE) to enable the codebook adjusting the different data or the model scales. Experiments on reconstruction shows the RVQ-VAE outperforms the VQ-VAE.

**Strengths:**

1-A new idea is proposed to extend the VQ-VAE to adjust the he different data or the model scales.

2-The experiments on the reconstruction shows its superority.

3-The paper is easy to follow.

**Weaknesses:**

1-Experiments on the reconstruction is not enough. The authors are suggested to included the experiemnts on 1-codebook usage, 2-image generation etc, to show the advantage of the method. Also the update strategy ( exponential moving average) of RVQ-VAE is also encouraged. I think this work is not thorough studied yet.

**Questions:**

1-I want to see thorough experiments of the proposed RVQ-VAE.

2-I am also not quiet sure the motivation of this work.

---

> ### Author Response · Authors · 2024-11-13
>
> Thank you for taking the time to review our work and for your valuable feedback. We appreciate your insightful comments and would like to address your concerns below.
>
>
> > Experiments on the reconstruction is not enough. The authors are suggested to included the experiemnts on 1-codebook usage, 2-image generation etc, to show the advantage of the method.
>
> Thank you for your sharp observation.
>
>  **Codebook Usage:** We included the codebook usage results  in Tables 8 and 9 of the Appendix. However, we omitted them from the main text because the trends were similar to those observed in the main experiment results. Specifically, when the codebook size is reduced below 256, we found that nearly all codebook entries are utilized. This indicates that smaller codebooks tend to have higher utilization rates, which aligns with our expectations.
>
> **Image Generation:** We agree that image generation is a major domain for VQ-based models and that evaluation in this area is essential. In our paper, we focused primarily on reconstruction to investigate whether rate adaptation could be achieved with minimal performance loss within the existing VQ framework. However, we acknowledge the importance of demonstrating the advantages of our method in image generation tasks as well.
>
> Unfortunately, due to an oversight in properly listing co-authors by the main author, we had to withdraw the paper. Nevertheless, we plan to include comprehensive image generation experiments in our next submission to better showcase the capabilities of our method.
>
> > Also the update strategy ( exponential moving average) of RVQ-VAE is also encouraged. I think this work is not thorough studied yet.
>
> Exponential Moving Average (EMA) is indeed a simple yet effective method for learning codebooks in VQ-VAE models. In our research, we applied EMA to the original codebook representations during the training process for both the model-based and data-driven approaches. Since we increased or reduced the codebooks based on these representations, EMA was effectively utilized in our method.
>
> We understand that this might not have been clearly written in the manuscript. To address concerns like yours, we will include a detailed explanation of the EMA update strategy in Section 3 of the revised paper.
>
> > I want to see thorough experiments of the proposed RVQ-VAE.
>
> We conducted experiments using baseline models such as VQ-VAE, VQ-VAE-2, and VQ-GAN to evaluate our proposed method. We agree that additional validation with more recent models like SQ-VAE is necessary. We are committed to including more extensive experiments in our next submission to provide a thorough evaluation of RAQ-VAE.
>
> > I am also not quiet sure the motivation of this work
>
> VQ-based generative models have emerged as a promising direction in the modern generative modeling field due to their ability to learn discrete representations of data, which can be efficiently used in various applications like image synthesis and compression. These models combine the strengths of variational autoencoders with vector quantization techniques to produce high-quality, diverse outputs.
>
> Our research team, including the authors, is particularly interested in applying these models to wireless communication scenarios. In such settings, it is highly desirable to have a high-efficiency framework that can serve multiple rates with a single VQ-based model, enabling devices to adapt to varying bandwidth conditions without the need for multiple models. This adaptability can significantly improve the scalability and efficiency of communication systems.
>
> We believe that our work addresses the scalability issues of VQ-based models by introducing rate adaptation capabilities. This not only contributes to the field of variable-rate image compression but also has practical implications for real-world applications where flexibility and efficiency are crucial.
>
>
> Thank you again for your thoughtful feedback. We appreciate your insights and will use them to improve our work in future submissions.

---

### Official Review · Reviewer_1Yqp · 2024-11-04

**Soundness:** 3
**Presentation:** 3
**Contribution:** 2
**Rating:** 5
**Confidence:** 4

**Summary:**

The paper introduces two codebook representation methods, model-based and data-driven approaches, which allow for varying the quantization rate within a single VQ-VAE model.

**Strengths:**

- RAQ-VAE allows a single model to adapt its codebook size without retraining to achieve variable rates for representations.
- RAQ-VAE has practical implications, particularly in applications that need adjustable compression rates such as real-time communications.

**Weaknesses:**

- The idea of rate adaptability is not new in the domain of neural compression, and the paper does not fully establish how RAQ-VAE advances beyond these existing frameworks. For instance, variable-rate neural image compression methods based on autoencoders and VAEs are prevalent, which is also mentioned in the related works.
- The experimental validation compares RAQ-VAE primarily to conventional VQ-VAEs. It is desirable to see if RAQ-VAE works with other methods, like more recent SQ-VAE and FSQ.
- The model-based approach is very weak.

**Questions:**

Why do we need the model-based approach?

---

> ### Author Response · Authors · 2024-11-13
>
> Thank you for your time reviewing and providing constructive feedback on our work. We would like to address your concerns below.
>
>
> > The idea of rate adaptability is not new in the domain of neural compression, and the paper does not fully establish how RAQ-VAE advances beyond these existing frameworks. For instance, variable-rate neural image compression methods based on autoencoders and VAEs are prevalent, which is also mentioned in the related works.
>
> We appreciate your insightful comments. As you have pointed out, our work is indeed related to the field of neural compression through rate adaptability, such as variable-rate neural image compression. In studies like QRes-VAE [A], the authors redesign their latent variable model using a quantization-aware posterior and prior, starting from a ResNet VAE. They present a powerful and efficient class of lossy image coders based on VQ methods that outperform other neural image compression techniques.
>
> Our research focuses on whether it is possible to achieve rate adaptation with minimal performance loss within the existing VQ framework. We believe that selecting a baseline VQ-VAE model optimized for variable-rate neural image compression and comparing its performance with other neural image compression methods is a valuable next step in our research.
>
> [A] Duan, Zhihao, et al. "Lossy Image Compression with Quantized Hierarchical VAEs." *Proceedings of the IEEE/CVF Winter Conference on Applications of Computer Vision*. 2023.
>
>
> > The experimental validation compares RAQ-VAE primarily to conventional VQ-VAEs. It is desirable to see if RAQ-VAE works with other methods, like more recent SQ-VAE and FSQ.
>
> We acknowledge that while we conducted experiments with baseline models such as VQ-VAE, VQ-VAE-2, and VQ-GAN, additional validation with recent methods like SQ-VAE and FSQ would strengthen our work. FSQ-VAE includes a unique quantization process for generating tokens used in MaskGIT, focusing on image-generative tasks rather than efficient data compression and reconstruction. FSQ and SQ-VAE have different quantization approaches, each with unique strengths in their respective applications. Therefore, determining whether rate-adaptive quantization is possible with FSQ-VAE requires further investigation, but comparison with SQ-VAE seems feasible. Unfortunately, due to an oversight in properly listing co-authors by the main author, we had to withdraw the paper. However, we plan to include these comparisons in our next submission opportunity.
>
> > Why do we need the model-based approach?
>
> We agree with your assessment that, in most cases, the model-based approach may perform worse than the data-driven approach. While the data-driven method increases training and inference overhead, it offers the advantage of achieving lower bitrates with minimal performance degradation. However, the model-based approach can be advantageous for practitioners with limited computational resources, as it only requires loading and applying codebook embeddings to a pre-trained VQ-VAE model without additional training. Despite its performance limitations, rate adaptation through the model-based approach could be another promising direction for future research. For instance, in large-scale models like ViT-VQGAN (Yu et al., 2022), which may experience significantly higher computational overhead compared to CNN-based models, this approach could be potentially useful.
>
> Once again, thank you for your valuable comments and suggestions. We will use this feedback to further improve our research.

---

### Note · Authors · 2024-11-15

**Comment:**

Dear Reviewers,

We would like to extend our heartfelt gratitude to each of you for taking the time to review our paper. Despite the unfortunate withdrawal of our submission due to an oversight in properly listing co-authors, we greatly appreciate your insightful comments and constructive feedback.

Your reviews have been invaluable in identifying areas for improvement, and we have made every effort to address your questions and concerns in our responses. Your contributions will significantly enhance the quality of our future work.

Thank you once again for your understanding and for the thoughtful consideration you have given to our research.

Sincerely, Submission6784 Authors.

**Withdrawal Confirmation:**

I have read and agree with the venue's withdrawal policy on behalf of myself and my co-authors.